# The systemic lupus erythematosus-associated NCF1⁹⁰ᴴ allele synergizes with viral infection to cause mouse lupus but also limits virus spread

Yanpeng Li [1,2], Ana Coelho [1], Zhilei Li [3], Malin Alsved [4], Qixing Li [2], Rui Xu [2], Huqiao Luo[1], Dongxia Liang [5], Jing Xu[6], Kutty Selva Nandakumar [2], Liesu Meng [5,6], Jakob Löndahl [4] & Rikard Holmdahl [1,2,5] ✉

Studying how single nucleotide polymorphisms (SNPs) crosstalk with non-autologous factors to cause complex autoimmune diseases is challenging. An amino acid replacement in the neutrophil cytosolic factor 1 (*NCF1*-339/ *NCF1^R90H*) leading to lower reactive oxygen species induction has been reported as the major SNP for systemic lupus erythematosus (SLE). Here we show that infection with the murine norovirus (MNV) contributes to the induction of lupus in *Ncf1⁹⁰ᴴ* mice. Mutant NCF1⁹⁰ᴴ upregulates the IFN-α/JAK1/STAT1 pathway in macrophages and anti-MNV-antibody production. In parallel, the MNV infection of NCF1⁹⁰ᴴ mice upregulates Toll-like receptor 7 in macrophages, plasmacytoid dendritic cells and B220⁺ splenocytes, thereby promoting germinal center formation and lupus-associated autoantibodies production. These compounded effects lead to protection against MNV infection but also glomerulonephritis with proteinuria and lupus arthritis in the absence of chemical inducers such as pristane. Our data thus suggest that this SLE-associated SNP, NCF1⁹⁰ᴴ, synergizes with MNV infection to induce the development of mouse lupus.

Systemic lupus erythematosus (SLE) is a common and complex autoimmune disease, caused by an interplay of many unknown genes and environmental factors. Yet, it has been difficult to precisely pinpoint and functionally confirm the causative factors. Thanks to genome-wide association studies (GWAS), hundreds of lupus disease-associated loci in the genome have been identified, which has strengthened the view that SLE is indeed a polygenic disease most likely regulated by variants in loci determining tolerance and immune responsiveness of the immune system[1]. However, the identification of the causative SNPs and the inducing environmental factors, remains largely unknown[2,3]. An alternative approach to identify the underlying genes has been to use genetic crosses of inbred animals susceptible to SLE. We have approached this problem by positioning the causative genes in mice and rats, and with these studies, we could identify some of the most important SNPs for complex autoimmune diseases.

[1]Medical Inflammation Research, Division of Immunology, Department of Medical Biochemistry and Biophysics, Karolinska Institute, Stockholm, Sweden. [2]SMU-KI United Medical Inflammation Center, School of Pharmaceutical Sciences, Southern Medical University, Guangzhou, China. [3]Clinical Pharmacy Division, Department of Pharmacy, Southern University of Science and Technology Hospital, Shenzhen, China. [4]Division of Ergonomics and Aerosol Technology, Faculty of Engineering, Lund University, Lund, Sweden. [5]National and Local Joint Engineering Research Center of Biodiagnosis and Biotherapy, Second Affiliated Hospital of Xi' an Jiaotong University (Xibei Hospital), Xi' an, China. [6]Key Laboratory of Environment and Genes Related to Diseases (Xi'an Jiaotong University), Ministry of Education, Xi'an, China. ✉e-mail: rikard.holmdahl@ki.se

The first autoimmune disease-associated gene cloned was the neutrophil cytosolic factor (*Ncf1*), encoding a protein (NCF1, alias p47phox) critically involved in the formation of the NADPH oxidase 2 (NOX2) complex, responsible for inducing reactive oxygen species (ROS) responses[4,5]. In rats, an amino acid replacing SNP (*Ncf1$^{T153M}$*) associated with autoimmune diseases was positioned[4,6], and an effect was reproduced in mice with a spontaneous *Ncf1$^{m1J}$* mutation[7]. In humans, the *NCF1* locus has been difficult to study due to copy number variations (CNV), but after exon sequencing a functionally important SNP could be identified[8]. This SNP (rs201802880 or *NCF1-339*, here denoted *NCF1$^{R90H}$*) replaces arginine with histidine at position 90 (R90H), affecting NCF1 interaction with the plasma cell membrane[5]. Analysis of SLE cohorts showed a high odds ratio (OR) and allelic frequency for *NCF1-339 (NCF1$^{90H}$)*[9,10]. Importantly, the NCF1$^{90H}$ allele caused a decreased ROS response in phorbol myristate acetate (PMA)-stimulated cells from patients with SLE in vitro, and it was also associated with an activated interferon signaling response, similar to previous findings in the *Ncf1$^{m1J}$* mutated mice[10–12]. BALB/c mice with the *Ncf1$^{m1J}$* mutation spontaneously showed signs of lupus with autoantibodies and glomerulonephritis and developed severe pristane-induced lupus, a commonly used mouse model of SLE[12,13]. The *Ncf1$^{m1J}$* mutation was associated with the activation of interferon signaling genes, including the STAT1 pathway[12]. An important remaining question is why and how the interferon response, and the development of lupus, are triggered in the presence of a low ROS-producing NCF1 variant. The detected antibodies to murine norovirus (MNV) in ROS-deficient *Ncf1$^{m1J}$* mutated mice with spontaneous lupus preliminarily suggested norovirus as a possible environmental factor[12]. In humans, there is no consensus on specific causative environmental challenges or virus infections, although DNA viruses, such as Epstein Barr virus (EBV), BK-virus (BKV), and retroviruses (RV) have been suggested to have disease association[14–16]. No substantial evidence for triggering lupus as a result of the severe acute respiratory syndrome coronavirus 2 (SARS-CoV-2) epidemic has been observed, although scattered cases have been reported. However, Covid-19 has been shown to be able to trigger the presentation or exacerbation of autoimmune diseases as well as a longstanding inflammatory syndrome in genetically predisposed patients[17–19].

Genetic predisposition accounts for approximately thirty percent of the risk of developing autoimmune diseases and interacting environmental factors could include toxic agents and diet but also infections[20]. The NCF1-regulated differences in the interferon signaling responses were seen in germ-free mice only after immunization, and they were also obvious after pristane injection, which triggered a more severe lupus-associated response[12,13,21]. Like several other mouse lupus models, pristane-induced lupus has been extensively studied for pathogenic mechanisms, and different downstream causative mechanisms have been suggested[22]. To understand the pathogenic role of the NCF1 polymorphism relevant to humans, we have addressed the role of the *Ncf1$^{R90H}$* alleles in mice.

Here we study the *Ncf1$^{R90H}$* alleles in mice, to understand the pathogenic role of the NCF1 polymorphism relevant to humans. To our surprise, the *Ncf1$^{90H}$* allele does not enhance pristane-induced lupus in the mice under specific pathogen-free conditions. However, infection with MNV causes the development of severe lupus in *Ncf1$^{90H}$* mice irrespective of whether these were naïve or injected with pristane. Thus, a synergy between a defined gene mutation and a specific non-autologous factor for the induction of an autoimmune disease is described.

## Results

### Environmental MNV induces lupus in BALB/c.*Ncf1$^{90H}$* mice

To determine lupus susceptibility, we followed C57BL/6NQ.*Ncf1$^{90H}$* and BALB/c.*Ncf1$^{90H}$* mice for 12 months, however, we did not find any signs of lupus. Thereafter, we injected them with pristane to establish the pristane-induced lupus (PIL) model but we did not see a significant

effect by the *Ncf1$^{90H}$* allele. However, years later after establishing the colony in our isolated SPF facility, we noticed a sudden outbreak of lupus in a cage of eight-week-old *Ncf1$^{90H}$* BALB/c mice, with arthritis, enlarged spleens, and proteinuria (Supplementary Figs. 1a–d). The only changed condition was a new virus infection and we found both antibodies and detection of live viruses, indicating a recent infection with murine norovirus (MNV) (Supplementary Table 1). We isolated virus RNA from mouse feces and confirmed the presence of a specific MNV strain 59591 by sequencing (Supplementary Table 2, Supplementary Fig. 2).

To confirm the causative role of MNV, we transferred the BALB/c.*Ncf1$^{R90H}$* littermates from an SPF facility to an isolated unit, and later added MNV-positive feces in the cages. We followed the mice for signs of virus infection, immune responses, and disease symptoms (Fig. 1a). Five weeks after MNV infection, *Ncf1$^{90H}$* mice started to develop lupus arthritis (Fig. 1b). Eight weeks later, proteinuria, and sera auto-antibodies against dsDNA, Sm/RNP, phospholipid (PLs), and antigens of β2-GP1 were increased in MNV-infected *Ncf1$^{90H}$* mice, compared to MNV-infected *Ncf1$^{R90}$* wild-type mice or non-MNV infected *Ncf1$^{90H}$* mice (Fig. 1c, d). As antibodies are important for clearing MNV infection[23] we monitored virus RNA, and anti-MNV antibodies. We found that the levels of virus RNA, 7 days post infection, were lower in the *Ncf1$^{90H}$* mice compared with *Ncf1$^{R90}$* mice. The levels of anti-MNV antibodies were positively correlated with the maximum arthritis score, levels of proteinuria, and anti-dsDNA antibodies in *Ncf1$^{90H}$* mice, with a trend of correlation with other investigated lupus-associated phenotypes (Fig. 1e, Supplementary Fig. 3a–c). Besides, we also noted a pronounced splenomegaly in MNV-infected *Ncf1$^{90H}$* mice, with an increased spleen index (Fig. 1f). Inflammatory cell infiltration was observed in the joint tissues of MNV-*Ncf1$^{90H}$* mice (Fig. 1g). Glomerulonephritis and interstitial mononuclear cell infiltrations were observed in the kidneys, with increased IgG and C3 deposits in the glomeruli (Fig. 1g, h). We found gene expression of *Ifnα* and anti-viral ISGs including *Irf7*, *Stat1*, *Mx1*, *Ip-10*, *and Isg15* in both spleens and kidneys, and also *Irf1* in the spleens to be increased in MNV infected *Ncf1$^{90H}$* mice, compared with wild-type *Ncf1$^{R90}$* (Fig. 1i, j). A pronounced phosphorylation of JAK1 and STAT1 was detected in the kidneys of MNV-*Ncf1$^{90H}$* mice compared to *Ncf1$^{R90}$* littermates (Fig. 1k, Supplementary Fig. 4a, b). We found decreased MNV load in the gastrointestinal tract, and sera antibodies clearing MNV displayed a positive correlation with lupus associated signs in the *Ncf1$^{90H}$* mice. These results suggest that the NCF1$^{90H}$ variant protects against MNV infection but allows the immune response against MNV to cause lupus.

### Environmental MNV aggravates pristane-induced lupus, with arthritis, in C57 black mice

We have previously shown that C57 black mice are susceptible to pristane-induced lupus by an NCF1-associated effect, mediated by hyperactive plasmacytoid dendritic cells (pDCs) by activating type I interferon and STAT1 pathways[24]. We found that MNV induces lupus in BALB/c.*Ncf1$^{90H}$* mice, and we investigated whether environmental factor could affect pristane-induced lupus (PIL) in B6N.Q.*Ncf1$^{90H}$* (BQ.*Ncf1$^{90H}$*) mice. Before that, we found an intraperitoneal injection of pristane in SPF-housed BQ.*Ncf1$^{90H}$* mice led to hyperactivated macrophages and neutrophils (Fig. 2a, b, Supplementary Fig. 5). MNV-infected homozygous female and male BQ.*Ncf1$^{90H}$* mice, and also heterozygous mice developed more severe lupus arthritis as compared with MNV-infected BQ.*Ncf1$^{R90}$* littermates in PIL (Fig. 2c, Supplementary Fig. 6a, b). The levels of anti-dsDNA antibodies were also increased in both female and male MNV-BQ.*Ncf1$^{90H}$* mice, compared with MNV-*Ncf1$^{R90}$* mice, seven months post-injection of pristane (Fig. 2d, Supplementary Fig. 6a–c). It demonstrates that MNV indiscriminately aggravates the diseases in both the female and male *Ncf1$^{90H}$* mice in the PIL model. We even detected slightly increased anti-dsDNA antibodies after MNV-infection, but before pristane injection, in *Ncf1$^{90H}$* compared with *Ncf1$^{R90}$*

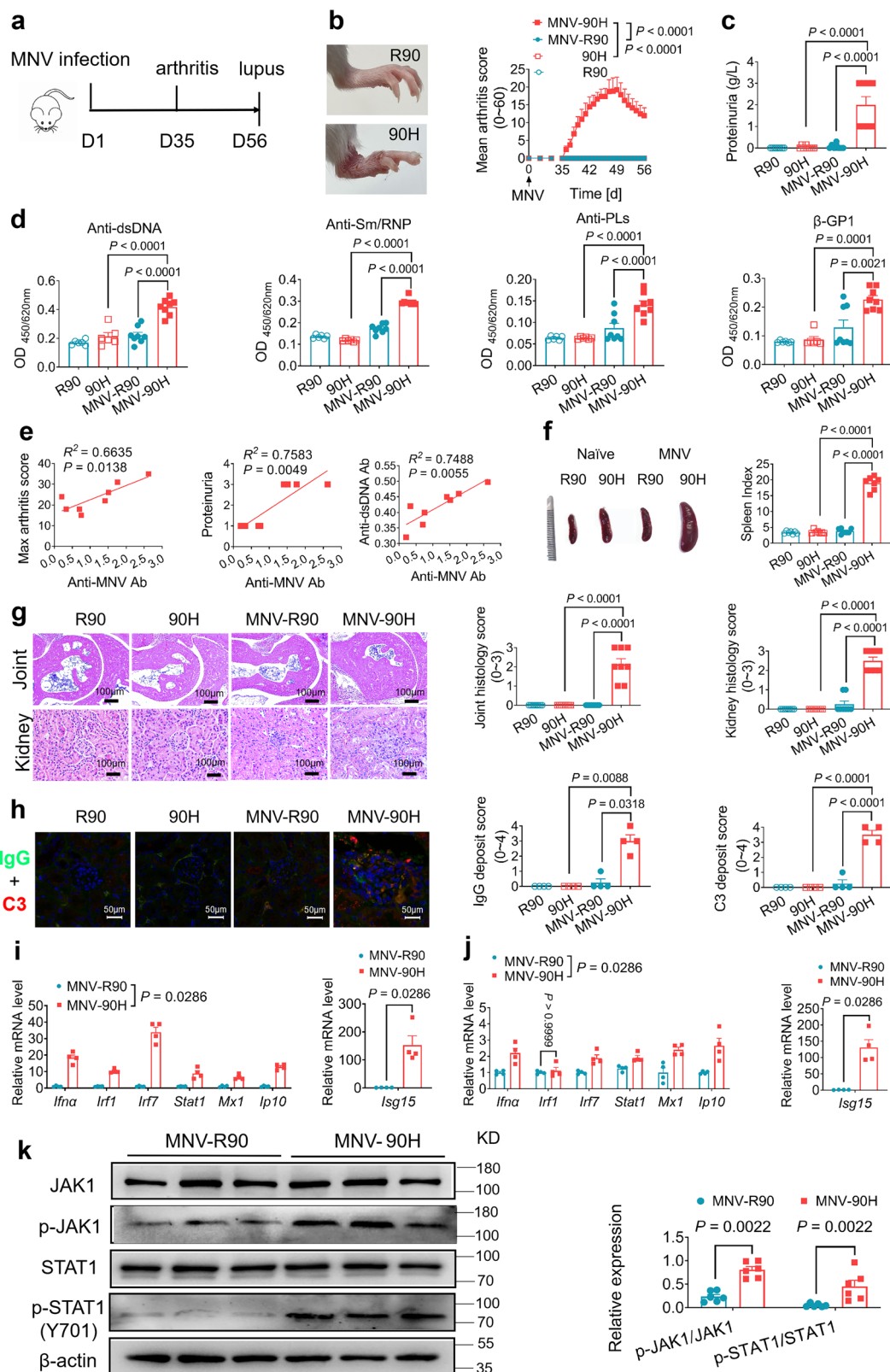

mice (Supplementary Fig. 6d). Anti-MNV antibodies were detected at seven-month post-injection of pristane, with and without MNV infection (Fig. 2e, Supplementary Fig. 6e). *Ncf1$^{90H}$* mice exhibited higher histology scores of joints, with more infiltrating cells, and higher histology scores of kidneys with increased mesangial stromal and cellularity, compared with wild-type mice seven months after pristane injection (Fig. 2f, Supplementary Fig. 6f, g). Given the above, we found that environmental

MNV infection enhanced the development of pristane-induced lupus in *Ncf1$^{90H}$* C57 black mice.

## The NCF1$^{90H}$ allele upregulates JAK1/STAT1 in mouse macrophages

Naïve SPF mice with the *Ncf1$^{90H}$* allele were healthy and did not differ in behavior or appearance compared to mice with the wild-type allele.

**Fig. 1 | Environmental MNV-infected _Ncf1⁹⁰ᴴ_ mice develop lupus. a** Timeline of the MNV infection experiment with male BALB/c. _Ncf1⁹⁰ᴴ_ mice after transferring from the SPF facility to the MNV-infected facility (day 1). **b** The representative appearance of arthritis in the ankle of hind paws on day 49 and mean arthritis scores from days 1 to 56 ($n = 8$ per group). Data were analyzed using one-way ANOVA and presented as mean ± SEM. **c** The levels of proteinuria ($n = 8$ per group). Data were analyzed using one-way ANOVA and presented as mean ± SEM. **d** The level of anti-dsDNA, anti-Sm/RNP, anti-phospholipid (anti-PLs) specific antibodies, and β2-GP1 on day 56 ($n = 8$ per group). Data were analyzed using one-way ANOVA and presented as mean ± SEM. **e** Correlation with anti-MNV antibodies (OD): maximum arthritis score ($p = 0.0138$), proteinuria ($p = 0.0049$), anti-dsDNA antibodies ($p = 0.0055$) in _Ncf1⁹⁰ᴴ_ mice on day 56 post of MNV infection ($n = 8$ per group). Data were analyzed using the Pearson correlation test. **f** Enlarged spleen and spleen index in mice ($n = 7$/group). The spleen index is defined by the spleen weight (mg) divided by the body weight (**g**) and then multiplied by 10 ($n = 8$ per group). Data were analyzed using one-way ANOVA and presented as mean ± SEM.

**g** Representative H&E stained joint and kidney sections (magnification x 10). Histological scoring of joint and kidney inflammation using a scale of 0-3 ($n = 8$ per group). Data were analyzed using one-way ANOVA and presented as mean ± SEM. **h** Immunofluorescence images (magnification x 20) and histology scores of deposits of IgG and C3 in the glomerulus from wild-type (WT) R90 and 90H mice on day 56 ($n = 4$ per group). Data were analyzed using one-way ANOVA and presented as mean ± SEM. **i** Relative expression of _Ifnα_ and anti-viral ISGs (_Irf1, Irf7, Stat1, Mx1, Ip10 and Isg15_) within spleens ($n = 4$ per group). Data were analyzed by the Mann-Whitney test (two-tailed) and presented as mean ± SEM. **j** Relative expression of _Ifnα_ and anti-viral ISGs (_Irf1, Irf7, Stat1, Mx1, Ip10 and Isg15_) within kidneys. The expression of mRNAs was normalized to the housekeeping gene β-actin ($n = 4$ per group). Data were analyzed by the Mann-Whitney test (two-tailed) and presented as mean ± SEM. **k** Immunoblot analysis of p-JAK1/JAK1 and p-STAT1/STAT1 proteins in the kidneys on day 56 after MNV infection ($n = 6$ per group). Data were analyzed by the Mann-Whitney test (two-tailed) and presented as mean ± SEM.

They also had normal _Ncf1_ mRNA expressions and normal numbers of mature macrophages (CD11b⁺ F4/80⁺ % Live CD45⁺) (Supplementary Fig. 7a, b). As expected, bone marrow-derived macrophages (BMDMs) from BQ._Ncf1⁹⁰ᴴ_ mice had lower intracellular and extracellular ROS production, as measured in naïve BMDMs or PMA-stimulated BMDMs. The NOX2 inhibitor GSK2795039 blocked both intracellular and extracellular ROS in _Ncf1⁹⁰ᴴ_ and _Ncf1ᴿ⁹⁰_ macrophages (Fig. 3a, b). To investigate functional downstream effects, we measured expression and phosphorylation of STAT1 (p-STAT1) in mouse macrophages, one of the key transcriptional factors activating ISG expression. Naïve macrophages from mice expressing NCF1⁹⁰ᴴ or with deficient NCF1 expression due to the _Ncf1ᵐ¹ᴶ_ mutation had a higher level of JAK1, STAT1, and STAT3 expression than wild-type BQ._Ncf1ᴿ⁹⁰_ mice. IFN-α treatment led to higher level phosphorylation of these molecules, which was more robust in _Ncf1⁹⁰ᴴ_ or _Ncf1ᵐ¹ᴶ_ macrophages (Fig. 3c–e) and confirmed by immunofluorescent staining (Supplementary Fig. 7c, d). NOX2 blocker treatment of macrophages from wild-type _Ncf1ᴿ⁹⁰_ mice enhanced IFN-α-induced phosphorylation of JAK1, STAT1, and STAT3, while $H_2O_2$ exposed macrophages from _Ncf1⁹⁰ᴴ_ and _Ncf1ᵐ¹ᴶ_ mice, had diminished phosphorylation (Fig. 3f–i). Investigation of the expression of STAT1 and p-STAT1 in peritoneal exudates macrophages (CD45⁺ CD11b⁺ F4/80⁺ Ly6C⁻) one day post-MNV intraperitoneal injection showed that p-STAT1 was upregulated in _Ncf1⁹⁰ᴴ_ mice compared to wild-type _Ncf1ᴿ⁹⁰_ (Fig. 3j, k). Taken together, the JAK/STAT signaling pathway in macrophages is profoundly activated by MNV infection in _Ncf1⁹⁰ᴴ_ mice due to low ROS production.

### The _Ncf1⁹⁰ᴴ_ allele protects against infection but induces lupus with an isolated MNV strain

To confirm the lupus-inducing effect of MNV and to understand its pathogenic function, we administered the MNV sub-strain Berlin/06/06DE S99 through oral gavage[25] (Fig. 4a). We could detect virus RNA in the small intestines and colons in all mice but not in the spleen, confirming that the MNV infection was localized to the gastrointestinal tract and did not spread systemically. Importantly, the levels in the small intestine and colon were dramatically lower in the BQ._Ncf1⁹⁰ᴴ_ mice compared with BQ._Ncf1ᴿ⁹⁰_ mice, measured seven days after MNV infection, showing that the lower capacity to produce ROS by NCF1⁹⁰ᴴ limits the infection (Fig. 4b). The BQ._Ncf1⁹⁰ᴴ_ mice, developed higher levels of IgG, compared to wild-type littermates, measured in sera 56 days after MNV infection, and induced an antibody response to MNV, in both males and females (Fig. 4c, d). The levels of anti-MNV antibodies in _Ncf1⁹⁰ᴴ_ mice correlated with the increase in total IgG levels (Fig. 4e. Supplementary Fig. 8a, b). The _Ncf1⁹⁰ᴴ_ mice had increased levels of autoantibodies against ssRNA, dsDNA and Sm/RNP, and also proteinuria (Fig. 4f–i). Based on the dramatically increased levels of IgG and higher titers of autoantibodies in MNV-infected _Ncf1⁹⁰ᴴ_ mice, we investigated the activation of T and B cells at the end of the experiment on day 84. The NCF1⁹⁰ᴴ allele did not affect the numbers of

T helper cells (Th) (CD4⁺) or regulatory T cells (Tregs) (CD4⁺ FOXP3⁺) cells among live CD45⁺ cells, but we found an increased expression of CD62L and CD69, indicating a higher activation (Supplementary Fig. 8c, d). The relative frequency of central or virtual memory T cells (Tcm/Tvm) (CD62L⁺ CD44⁺) in CD4⁺ Foxp3⁺, T follicular cells (Tfh) (CXCR5⁺ PD-1⁺) among CD4⁺ FOXP3⁻ cells and T follicular regulatory (Tfr) (CXCR5⁺ PD-1⁺) among CD4⁺ Foxp3⁺ cells, were increased whereas tissue-resident memory T cells (Trm) (CD62L⁻ CD69ʰⁱ) among CD4⁺ CD44ʰⁱ cells, were decreased in Peyer's patches (PPs) of _Ncf1⁹⁰ᴴ_ mice (Fig. 4j–m, Supplementary Fig. 8e). Instead, the number of Tfh and Tfr cells were not changed in the small intestine outside PPs (Supplementary Fig. 8f–h). Both Tfh and Tfr cells regulate the generation of antigen-specific antibody-secreting cells (ASCs) in the germinal center (GC). The population of ASCs (IgDˡᵒ CD138⁺ Sca-1⁺) from PPs or the small intestine was not changed (Supplementary Fig. 8f, i, j). Analysis of the B cell compartment showed that GC-B cells were increased in the PPs of _Ncf1⁹⁰ᴴ_ mice, indicating a strong B cell activation in response to the MNV infection (Fig. 4n, Supplementary Fig. 8f). We found an increase of long-lived plasma cells (LLPCs) in the PPs of _Ncf1⁹⁰ᴴ_ mice (Fig. 4o, Supplementary Fig. 8f, k). We conclude that the MNV gastrointestinal infection in _Ncf1⁹⁰ᴴ_ mice trigger a strong activation of T and B cells, both in the intestine and systemically in germinal centers, leading to the expansion of LLPCs.

### Non-mucosal MNV infection induces lupus arthritis in _Ncf1⁹⁰ᴴ_ mice

The virus oral infection was localized to the gastrointestinal tract but also triggered the immune responses in draining immune organs like the PPs and mesenteric lymph nodes (mLNs). Importantly, it also activated the immune system systemically, as indicated by a strong response in the spleen. We were curious to see if the virus could trigger a similar immune response by extra-intestinal exposure and thereby induce lupus. We therefore infected the mice with the MNV strain intravenously and intraperitoneally (Fig. 5a). An immune response to the MNV developed in both groups of mice, as seen by an induced antibody response to MNV (Fig. 5b). Similar to the per orally infected mice, we found dramatically increased titers of anti-ssRNA autoantibodies and production of IgG in mice with the _Ncf1⁹⁰ᴴ_ allele (Fig. 5c, d). All BQ._Ncf1⁹⁰ᴴ_ mice developed arthritis, but none of the wild-type BQ._Ncf1ᴿ⁹⁰_ mice, with onset as early as day 14, reaching maximum severity on day 35 (Fig. 5e–h). We also measured the levels of antibodies to type II collagen (COL2) since these are prominent natural autoantibodies in mouse lupus but may also occur as a response to joint inflammation[26,27]. Mild but significant increased IgG, IgG2b, and IgM antibodies to COL2 were detected in MNV-_Ncf1⁹⁰ᴴ_ mice on day 21 compared with naive _Ncf1⁹⁰ᴴ_ mice (Fig. 5i–k). B cell-ELISpot data showed consistent results of increased numbers of anti-COL2 IgG and IgM antibody secreting cells (ASCs) in the spleen of _Ncf1⁹⁰ᴴ_ mice after injection of MNV when compared with naive mice or MNV-_Ncf1ᴿ⁹⁰_

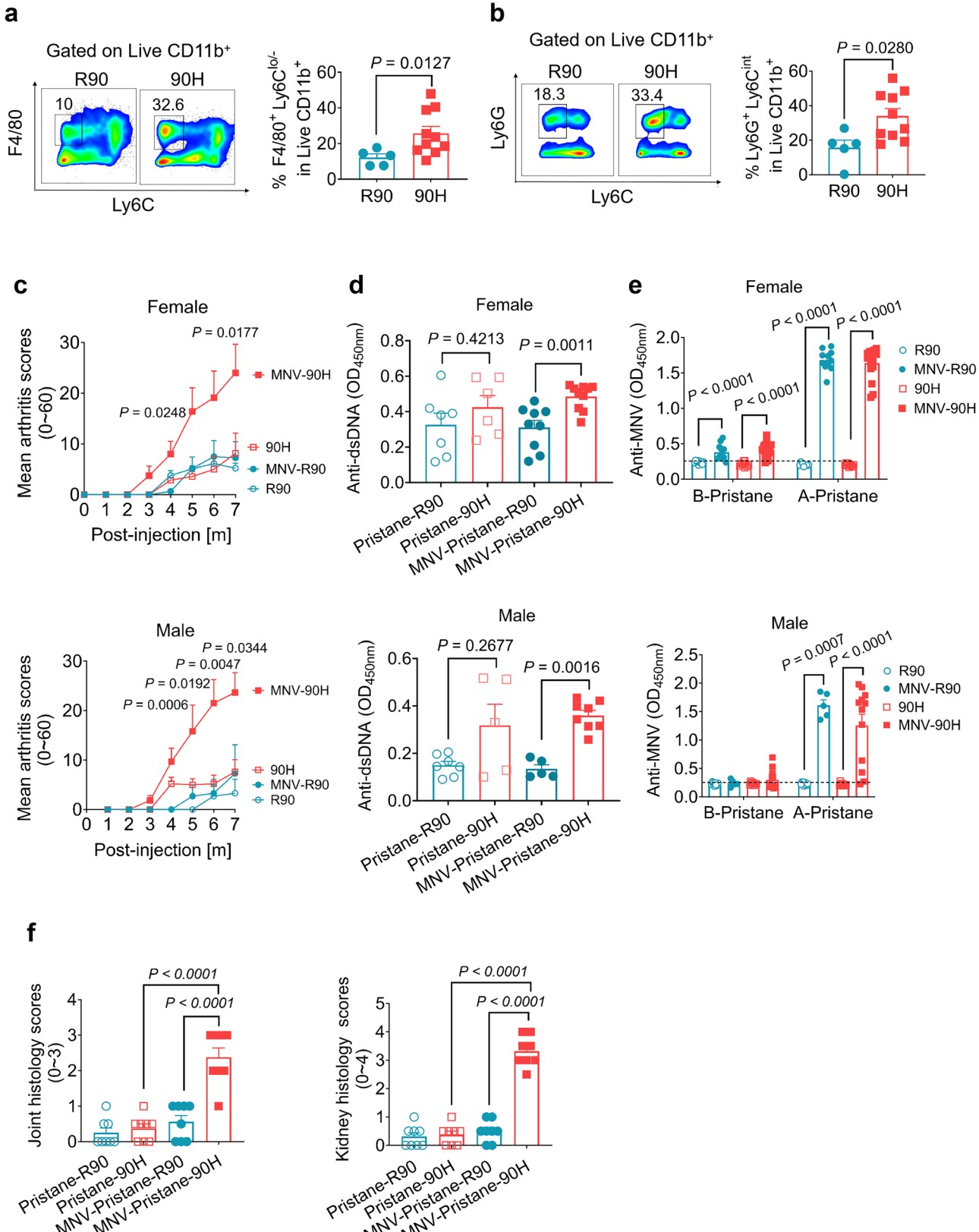

mice (Fig. 5l, m). In the late stage, on day 56, we also found increased anti-dsDNA and anti-Sm/RNP antibodies, as well as proteinuria in MNV-injected mice compared to naïve mice (Supplementary Fig. 9). Taken together, we showed that extraintestinal injection of MNV trigger a more rapid activation of the immune system, leading to lupus arthritis and induction of typical lupus autoantibodies in *Ncf1*[90H] mice.

## NCF1[90H] allows development of lupus through T cell-dependent germinal center response

We could confirm that the natural peroral MNV infection was restricted locally to the gastrointestinal tract, but in BQ.*Ncf1*[90H] mice the virus infection was more restricted. However, it distorted the systemic immune response driving it to convert to pathogenic autoimmunity.

**Fig. 2 | MNV aggravates pristane-induced lupus in BQ.***Ncf1*[90H] **mice.** Gating and population of immune cells in the peritoneal cavity of the male 90H mice 3 days after pristane injection. **a** Macrophages (CD11b⁺ F4/80⁺ Ly6C⁻) (R90: $n = 5$; 90H: $n = 10$). Data were analyzed by the Mann-Whitney test (two-tailed) and presented as mean ± SEM. **b** Neutrophils (CD11b⁺ Ly6G⁺ Ly6C⁻) (R90: $n = 5$; 90H; $n = 10$). Data were analyzed by the Mann-Whitney test (two-tailed) and presented as mean ± SEM. **c** Mean arthritis scores before and after pristane injection (0–7 months) in female (R90: $n = 9$; 90H: $n = 7$; MNV-R90: $n = 12$; MNV-90H: $n = 8$) and male (R90: $n = 7$; 90H: $n = 5$; MNV-R90: $n = 7$; MNV-90H: $n = 6$) mice, with and without MNV infection. MNV-R90 vs MNV-90H. Data were analyzed by the Mann-Whitney test (two-tailed) and presented as mean ± SEM. **d** Levels of anti-dsDNA antibodies in non-MNV infected female mice (Pristane-R90: $n = 7$ and Pristane-90H: $n = 6$) and male mice

(Pristane-R90: $n = 7$ and Pristane-90H: $n = 5$) and MNV infected female mice (MNV-Pristane-R90: $n = 9$ and Pristane-90H: $n = 10$) and male mice (MNV-Pristane-R90: $n = 5$ and MNV-Pristane-90H: $n = 8$) seven months post-injection of pristane. Data were analyzed by the Mann-Whitney test (two-tailed) and presented as mean ± SEM. **e** Levels of anti-MNV antibodies in non-MNV infected female mice (Pristane-R90: $n = 10$ and Pristane-90H: $n = 10$) and male mice (Pristane-R90: $n = 10$ and Pristane-90H: $n = 10$) and MNV-infected female mice (MNV-Pristane-R90: $n = 11$ and MNV-Pristane-90H: $n = 17$) and male mice (MNV-Pristane-R90: $n = 5$ and MNV-Pristane-90H: $n = 12$). Data were analyzed by the Mann-Whitney test (two-tailed) and presented as mean ± SEM. **f** Statistics of histology scores of joints and kidneys seven months post-injection of pristane ($n = 8$ per group). Data were analyzed using one-way ANOVA and presented as mean ± SEM.

---

Therefore, we focused on the maturation and differentiation of B and T cells in the spleen of the *Ncf1*[90H] mice. Already in naïve *Ncf1*[90H] mice, we found an increased population of mature B cells (CD93⁻, gated on CD19⁺ B220⁺), but not B2 cells, along with an increased number of follicular B (FOB) cells (CD23⁺ CD21⁺, gated on CD19⁺ B220⁺ CD93⁻), and naive FOB cells (IgD^hi IgM^lo, gated on CD23⁺ CD21⁺ CD19⁺ B220⁺ CD93⁻). Only the ratio of marginal zone B cells (MZB) in live cells from *Ncf1*[90H] mice showed an increase, and not the IgD⁻ IgM^hi MZB cells (Fig. 6a–c, Supplementary Fig. 10a–d).

Next, we investigated the effect of MNV infection on the B cells and T cells in the spleen. We found that the population of naïve B2 cells (IgD^hi IgM^lo B2 cells) was increased by the MNV infection but the increase was not influenced by the genotype (Fig. 6d, Supplementary Fig. 10e). However, the expression of MHCII and CXCR4 on B2 cells were increased in *Ncf1*[90H] mice (Fig. 6e). MNV infection led to a decrease of regulatory T cells (% CD4⁺ FOXP3⁺ in live cells), but an increase of CD44 expression on Tregs in *Ncf1*[90H] mice (Supplementary Fig. 10f, g). In the *Ncf1*[90H] mice, MNV infection reduced the number of Th cells, but with an increased expression of PD-1, CXCR5, CD44, and CD69 (Fig. 6f, Supplementary Fig. 10h, i). Tem cells (CD44⁺ CD62L⁻, gated on CD4⁺ FOXP3⁺) were increased by MNV infection in *Ncf1*[90H] mice, whereas naive Tregs (CD44⁻ CD62L⁺, gated on CD4⁺ FOXP3⁺) and Tcm/ Tvm cells (CD62L⁺ CD44⁺, gated on CD4⁺ FOXP3⁺) were decreased (Fig. 6g). MNV infection also led to an expansion of follicular T cells, in particular in *Ncf1*[90H] mice, in both Tfh (CXCR5⁺ PD-1⁺, gated on CD4⁺ FOXP3⁻) and Tfr (CXCR5⁺ PD-1⁺, gated on CD4⁺ FOXP3⁺) compartments (Fig. 6h, i). Similarly, the frequency of Trm (CD62L⁻ CD69^hi, gated on CD4⁺ CD44^hi) were increased by MNV infection in *Ncf1*[90H] mice (Fig. 6j). Considering the importance of the GC response in lupus, we explored the impact of NCF1-mediated ROS burst on GC-B cells. We found a dramatically increased population of GC-B cells (CD38^lo GL7⁺ cells, gated on B220⁺ CD19⁺ IgD⁻ population) in *Ncf1*[90H] mice, compared to wild-type *Ncf1*[R90] or non-MNV injected *Ncf1*[90H] mice (Fig. 6k, Supplementary Fig. 10j). We also measured ASC (IgD^lo CD138⁺ Sca-1⁺) sub-populations including PBs (CD19⁺ B220⁺), newly formed PCs (NEW PCs) (CD19⁺ B220⁻), and LLPCs (CD19⁻ B220⁻). Importantly, the number of ASCs was increased in *Ncf1*[90H] mice after MNV infection (Supplementary Fig. 10k). We also observed an increased population of PBs, NEW PCs, and LLPCs in MNV-infected *Ncf1*[90H] mice, compared with MNV-infected wildtypes and also increased numbers of LLPCs in MNV-*Ncf1*[90H] mice, compared to naive *Ncf1*[90H] mice (Fig. 6l). Taken together, the *Ncf1*[90H] allele promoted maturation and differentiation of B and T cells, and the NCF1[90H] allele, in response to MNV infection, mediated a T cell-dependent GC response.

**The *Ncf1*[90H] allele promotes the upregulation of TLR7**
TLR7 is thought to localize in endolysosomes and are also present on the cell surface of immune cells, with responses in dendritic cells, macrophages and B cells[28]. A translocation from the X chromosome onto the Y chromosome causes overexpression of the translocated genes, which includes the ssRNA-recognizing TLR7 member of the TLR family of receptors in male mice bearing the y-linked autoimmune

accelerating (*Yaa*) locus, which is sufficient to enhance TLR7-mediated activation of innate immune responses and lupus development[29–31]. To investigate the role of TLR7 in ssRNA virus- MNV-induced lupus. We took advantage of the Yaa locus and confirmed that intact TLR7 is upregulated in the spleen of naïve *Ncf1*[90H] mice with the *Yaa* locus (Fig. 7a). We also found expression of intact TLR7 to be increased in environmental MNV-infected homozygous *Ncf1*[90H] mice with *Yaa* locus (Fig. 7b). In littermate mice, without the *Yaa* locus, non-mucosal MNV infection had an increased ratio of cleaved TLRs to intact TLRs and increased levels of intact TLR7 (Fig. 7c). Similarly, mucosal-MNV infection upregulated intact TLR7 expression in *Ncf1*[90H] mice (Fig. 7d). We did not observe the effect by *Yaa* locus on the ratio of immune cells but we found dramatically increased populations of macrophages (CD45⁺ CD11b⁺ F4/80⁺) and pDCs (CD11c^lo/int Ly6C^hi PDCA-1⁺ B220⁺) in MNV infected spleens, compared to spleen without infections, in particular after the mucosal infection (Supplementary Fig. 11 and 12a, b). TLR7 is expressed in multiple immune cell subsets, including B cells, cDCs, pDCs, macrophages, and monocytes. Based on this, we investigated the TLR7 and TLR9 expressions on macrophages, pDCs, and B220⁺ splenocytes. It displayed an increased level of TLR7 in macrophages, pDCs, and B220⁺ cells from naïve *Ncf1*[90H] mice with the *Yaa* locus, compared to mice without the *Yaa* locus (Fig. 7e–g). TLR9 was not changed in macrophages, pDCs, or B220⁺ cells by the *Yaa* locus or the NCF1[90H] variant (Supplementary Fig. 11 and 12c–e). TLR7 expression was increased in macrophages, pDCs, and B220⁺ splenocytes after mucosal MNV infection in *Ncf1*[90H] mice. An increase only in macrophages by non-mucosal infection was observed in *Ncf1*[90H] mice. In addition, it showed a dramatic increase of TLR7 in macrophages and B220⁺ cells by mucosal MNV infection compared with non-mucosal infection (Fig. 7h–j). In contrast, the *Ncf1*[90H] allele had no significant effect on TLR9 expression with or without MNV infection (Supplementary Fig. 12f–h). Taken together, the NCF1[90H] variant upregulates the MNV-induced TLR7 signaling pathway, to mediate activation of innate immune responses and lupus development.

## Discussion
We have observed that infection with MNV, a normally non-pathogenic virus, triggers a disease mimicking lupus in mice with SNP associated with SLE in humans. On the other hand, the lupus causative SNP leading to a lower ROS response, protected the mice from the virus infection by restricting the spread of the virus. MNV is a single-stranded RNA virus infecting intestinal epithelial cells but which is not spreading systemically. Importantly, it activates not only the local immune response but also the immune system systemically leading to activation of both T and B cells with enlargement of spleens. Upon infection in low ROS-producing *Ncf1*[90H] mice, the virus releases ssRNA, triggering the interferon signaling pathway, possibly through activation of TLR7 and the STING pathway[32,33]. NCF1 is expressed mainly in phagocytic and antigen-presenting cells, including macrophages, dendritic cells, and B cells. Activation of these cells triggers the NOX2 complex to produce ROS. And the downstream oxidative metabolites such as hydrogen peroxide will oxidize regulatory cysteines in many

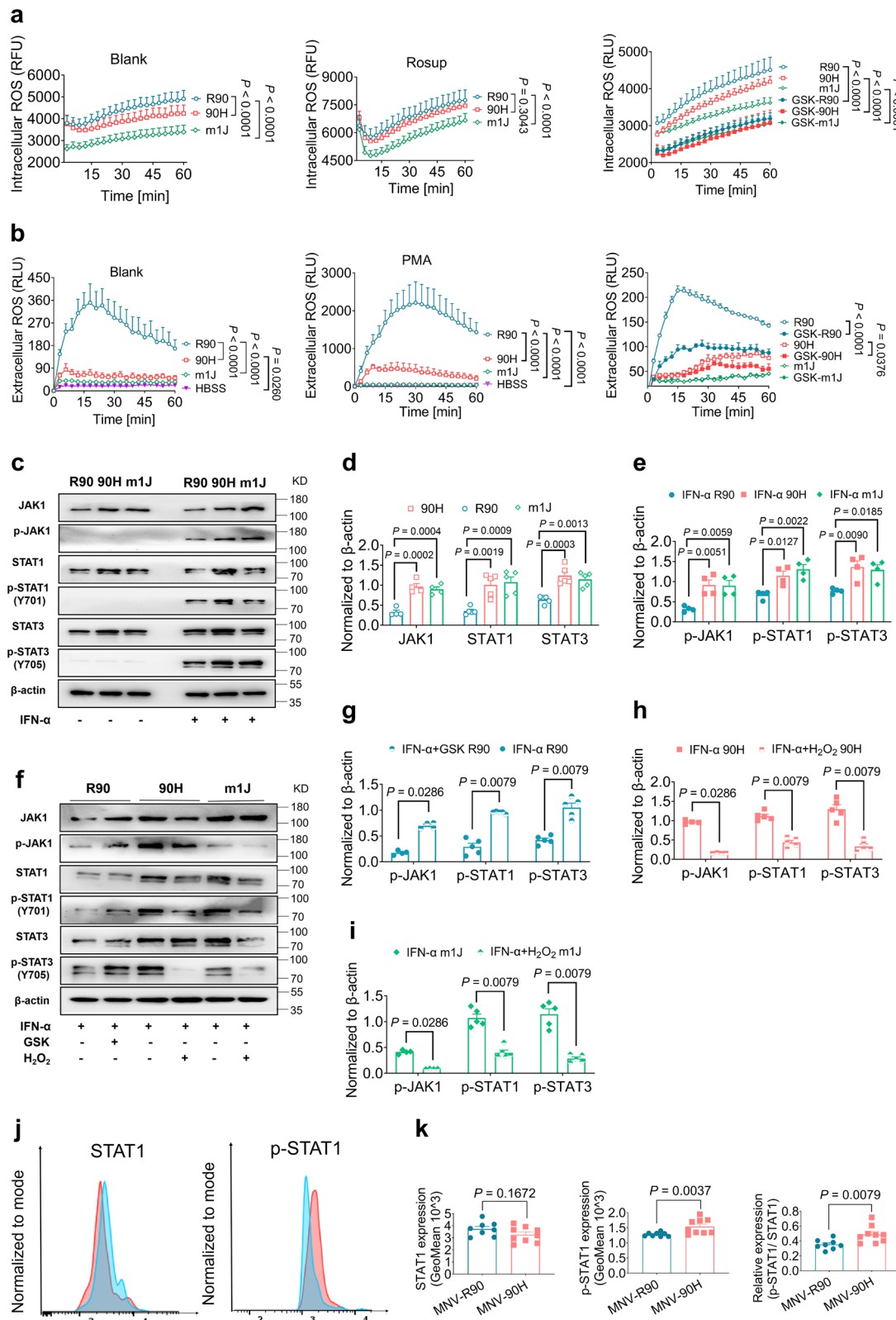

proteins intracellularly and protect the cell from an exhaustive activation of JAK/ STAT mediated activation of interferon signaling[34]. However, a NOX2 complex with a lower capacity to produce ROS, due to *NCF1^90H*, will allow phosphorylation of STAT1 and activate the interferon signaling cascade. We have earlier shown that the critical target cell for the development of mouse lupus is the plasmacytoid dendritic cell, which downstream lead to B cell activation and

autoantibody production[24]. From the perspective of immune defense against viruses, it makes sense as it will lead to more efficient virus neutralization. A genetically determined lower capacity to produce ROS from the NOX2 complex could therefore be beneficial for the protection of virus infection, particularly ssRNA viruses, known to efficiently activate interferon signaling through TLR7 or STING. However, if the infection is prolonged and turns into chronic inflammation,

**Fig. 3 | The NCF1⁹⁰ᴴ allele upregulates IFN-α/JAK1/STAT1 pathway in mouse macrophages. a** Intracellular ROS in BMDMs without (Blank) and with (Rosup) stimulation ($n = 8$ per group). Intracellular ROS in BMDMs with NOX2 inhibitor GSK2795039 ($n = 5$ per group). R90 vs 90H, or m1J. Data were analyzed using one-way ANOVA and presented as mean ± SEM. R90 vs GSK-R90, 90H vs GSK-90H, or m1J vs GSK-m1J, Data were analyzed by the Mann-Whitney test (two-tailed) and presented as mean ± SEM. **b** Extracellular ROS production in BMDMs without (Blank) and with (PMA) stimulation ($n = 5$ per group). Extracellular ROS in BMDMs after treatment with NOX2 inhibitor GSK2795039 ($n = 3$ per group). R90 vs GSK-R90, 90H vs GSK-90H, m1J vs GSK-m1J. R90 vs 90H or m1J, 90H vs Hank's Balanced Salt Solution (HBSS). Data were analyzed using one-way ANOVA and presented as mean ± SEM. R90 vs GSK-R90, 90H vs GSK-90H, or m1J vs GSK-m1J, Data were analyzed by the Mann-Whitney test (two-tailed) and presented as mean ± SEM. **c** Western blots of JAK1, STAT1, and STAT3 proteins in naïve macrophages, and after IFN-α stimulation from 90H and m1J male mice, compared to wild-type (R90) male mice ($n = 5$ per group). **d, e** Quantification of relative protein expression, normalized to β-actin ($n = 4$ per group). Quantitative comparisons between samples on the

same blots or the samples derive from the same experiment and that blots were processed in parallel. R90 vs 90H, R90 vs m1J. IFN-α R90 vs IFN-α 90H, IFN-α R90 vs IFN-α m1J. Data were analyzed using one-way ANOVA and presented as mean ± SEM. **f** Western blots of p-JAK1, p-STAT1, and p-STAT3 in IFN-α stimulated macrophages, together with adding *Ncf1*/NOX2 blocker to cells from wild-type mice ($n = 5$ per group), or adding $H_2O_2$ to cells from 90H or m1J mice ($n = 5$ per group). **g–i** Quantification of relative protein expression, normalized to β-actin ($n = 5$ per group). Data were analyzed by the Mann-Whitney test (two-tailed) and presented as mean ± SEM. **j** Representative histograms of STAT1 and p-STAT1 in peritoneal exudates macrophages (CD45⁺ CD11b⁺ F4/80⁺ Ly6C⁻) one- day post-MNV injection (1dpi) by IP. **k** Geometric mean of STAT1 and p-STAT1, and relative expression of p-STAT1/STAT1 within peritoneal exudates macrophages (MNV-R90: female: $n = 4$, male: $n = 4$; MNV-90H: female: $n = 5$, male: $n = 4$). Data were analyzed by the Mann-Whitney test (two-tailed) and presented as mean ± SEM. BMDMs were obtained from the differentiation of monocytes recovered from the femur and tibia of 6 to 8- week- old male B6N.Q.*Ncf1*ᴿ⁹⁰, B6N.Q. *Ncf1*⁹⁰ᴴ, B6N.Q and B6N.Q.*Ncf1*ᵐ¹ᴶ mice.

the lack of ROS allows an exaggerated immune activation, which may lead to systemic autoimmunity.

The SNP coding for the *NCF1*⁹⁰ᴴ variant was detected by exon sequences of NCF1 copies in humans. Targeted mutations leading to *Ncf1*ᴿ⁹⁰ᴴ replacement in mice have independently been made by several groups in parallel[24,35–37]. The disease-causative effect in humans has been partly reproduced based on the commonly used protocol of pristane or imiquimod-induced lupus but with the suggestion of different downstream mechanisms. In one report, mild spontaneous lupus developed in aged (>5 months) B6 mice but could be exacerbated after pristane injection, and the disease development was dependent on decreased efferocytosis leading to an expansion of Tfh2 cells[35]. In another study, the *Ncf1*⁹⁰ᴴ mice did not develop spontaneous lupus but had severe disease after imiquimod (IMQ) treatment[36]. Clearly, the induced disease manifestations associated with the NCF1⁹⁰ᴴ variant are dependent on the environmental trigger. For example, intraperitoneal injection of mannan induces severe psoriasis and psoriatic arthritis[37]. Acidification of phagosomes, due to low ROS in phagosomes of macrophages and pDCs, has been suggested as the initial effect[35,36]. On the other hand, we have shown that NCF1 deficiency due to the *Ncf1*ᵐ¹ᴶ mutation modifies regulation of cysteine oxidation and could thereby affect different pathways such as interferon signaling, activation of antigen receptors, or affecting antigen processing[12,34,38–40].

Mutations in different NOX2 components have been shown to enhance various autoimmune diseases, but with different downstream mechanisms[13,41–43]. Symptoms mimicking lupus, with autoantibodies and glomerulonephritis, appeared spontaneously in naïve ROS-deficient BALB/c. *Ncf1*ᵐ¹ᴶ mutated mice, which could be explained as mice in the SPF animal house for these experiments were exposed to recurrent MNV infections as they had antibodies to MNV[12]. Pristane injection has been shown to induce mild lupus in C57 black mice even in an MNV-free SPF facility, and the disease was enhanced by the *Ncf1*ᵐ¹ᴶ mutation[24]. In comparison with the *Ncf1*ᵐ¹ᴶ mutation, the *Ncf1*⁹⁰ᴴ mutation has more remaining ROS-inducing capacity. In our study, pristane-injection did not induce significant signs of lupus, or lupus-associated antibodies, in *Ncf1*⁹⁰ᴴ mice neither having C57 black nor BALB/c genes predominantly. Instead, significantly increased lupus diseases only developed after MNV infection, regardless of whether they had been injected with pristane or not, and the development of the disease with MNV infection was completely dependent on the NCF1⁹⁰ᴴ allele. Thus, as compared with pristane injections, it is possible that MNV triggers a different mechanism signaling pathway leading to lupus symptoms.

Since MNV is normally not a pathogenic virus in common mouse strains, it has previously not been included in many SPF protocols and its importance was identified only recently. There are different

substrains of MNV and here is shown that two different substrains can induce lupus, however, it might not apply for all substrains. MNV is known to primarily affect the intestinal epithelial cells, similar to the human norovirus, but it is unclear whether the infection spreads systemically, and infecting macrophages, dendritic cells, and B cells in vivo[44]. It was earlier reported that noroviruses can infect B cells and impair B cell development within the bone marrow in a STAT1-dependent but IFN signaling-independent manner[45]. Type I IFN (IFN-α/β) controls systemic replication, and type III IFN (IFN-λ) controls MNV persistence in the intestinal epithelium[31,46,47]. In our work, we clarified that MNV infection triggers TLR7 and activates IFN signaling in the background of ROS deficiency. The NCF1⁹⁰ᴴ variant could activate IFN signaling by several mechanisms, including lowering endosomal pH and direct inhibition of oxidation in the regulatory cysteines present in JAK1 and STAT1. This work is compatible with that the virus infects the intestinal epithelia, including macrophages and dendritic cells surveilling incoming infections, which in turn activates the immune system in PPs, draining lymph nodes, and subsequently the systemic immune system.

The key event is the exposure of viral ssRNA, which under reducing conditions activates interferon signaling pathways leading to a strong activation of antigen presenting cells including B cells, and subsequently autoreactive T cells. This pathogenic event is possible due to an interaction between a commonly occurring cytosolic protein, NCF1⁹⁰ᴴ, through the formation of the NOX2 complex, and a commonly occurring environmental factor, an ssRNA virus. The corresponding human norovirus-induced infections are widely spread, but transient and asymptomatic in most individuals and an association with SLE is therefore difficult to determine[48]. Both the structure and function of NCF1 and the NOX2 complex are, however, highly conserved.

Most importantly, even the functional role of the NCF1 polymorphism is similar between rodents and humans. It is therefore likely that a functional ROS response by the NOX2 complex gives relative protection against norovirus infections, but also an increased risk of developing SLE and other autoimmune disorders. Our discovery of MNV as a trigger of lupus in *Ncf1*⁹⁰ᴴ mice does not exclude other viruses, environmental factors or genes, to be causative for SLE. To the best of our knowledge, this is a disease-causative interaction has been shown between a defined genetic and a specific environmental factor to control a complex autoimmune disease.

## Methods
### Construction of mice
Balb/cByJ (stock 001026) and C57BL/6NJ (stock 005304) were originally from The Jackson Laboratory. B6N.Q (C57/B6N.Q/rhd) and B10.Q (C57/B10N.Q/rhd) have been fully backcrossed into B6N and B10

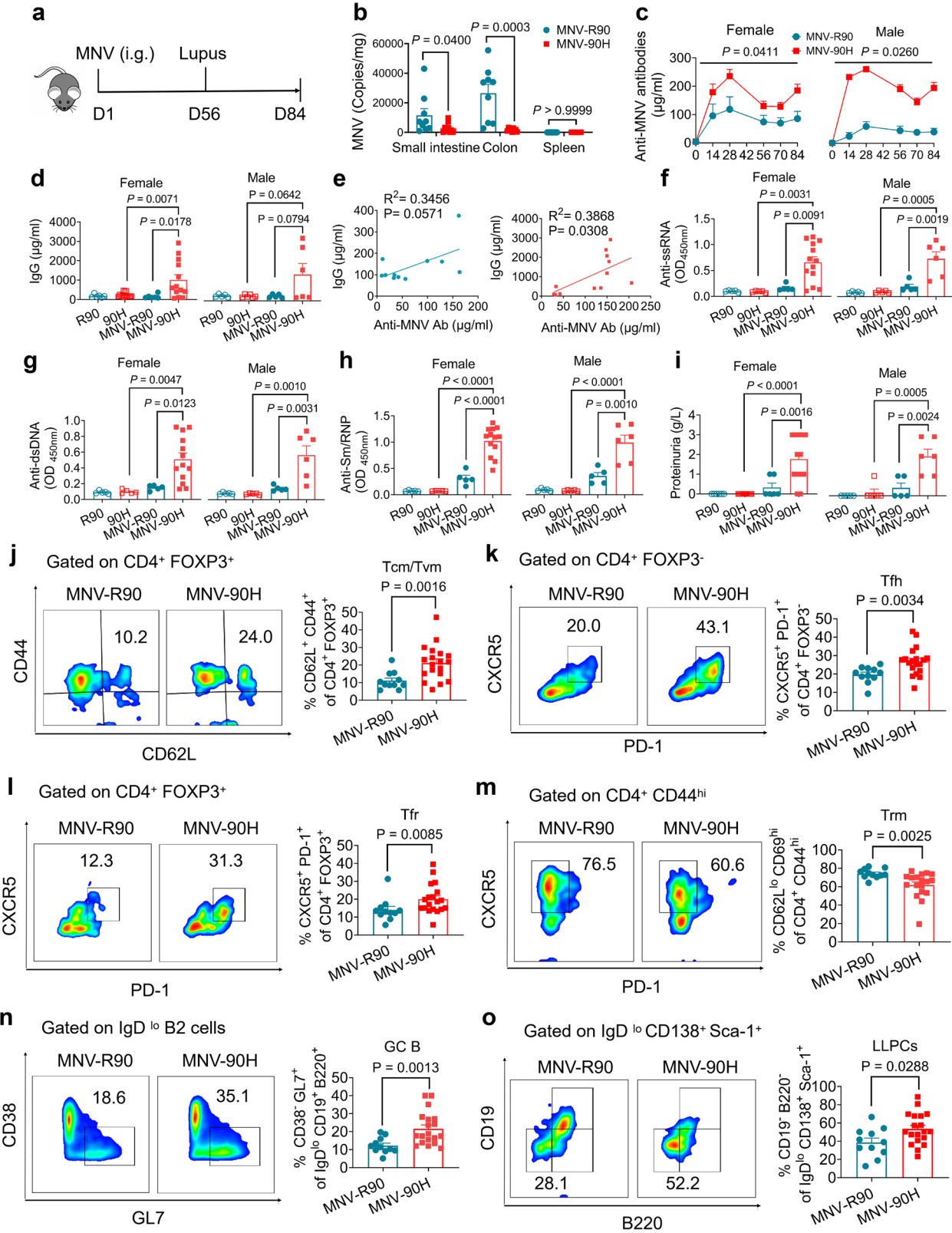

genomes but with an MHCII A$^q$ containing haplotype from DBA/1 to establish the BQ.*Ncf1$^{90H}$* and BQ.*Ncf1$^{m1J}$* strains[37,49]. The backcrossing onto the BALB/c to establish the BALB/c.*Ncf1$^{90H}$* strain. Age- and sex-matched littermates from heterozygous intercrosses were used in all experiments. For MNV-infected experiments in vivo, *Ncf1$^{90H}$* (abbreviated as 90H) refers to BALB/c.*Ncf1$^{90H/90H}$*, *Ncf1$^{R90H}$* (abbreviated as R90H) refers to BALB/c.*Ncf1$^{R90/90H}$* and *Ncf1$^{R90}$* (abbreviated as R90)

refers to BALB/c.*Ncf1$^{R90/R90}$* mice. Unless otherwise stated, *Ncf1$^{90H}$* (abbreviated as 90H) refers to B10.Q.*Ncf1$^{90H/90H}$* and *Ncf1$^{m1J}$* (abbreviated as *m1J*) refers to B10.Q.*Ncf1$^{m1J/m1J}$* mice. B6.SB-*Yaa*/J (stock 000483) from the Jackson Laboratory has been fully backcrossed onto B10.Q, denoted as B10.Q.*Yaa*. The *Yaa*-carrying strain with fully functional NCF1 (R90.*Yaa*) or with human *NCF1$^{90H}$* allele (90H.*Yaa*) was obtained by crossing BQ.*Ncf1$^{90H/90H}$* with B10.Q.*Yaa*. The mice were

**Fig. 4 | Mucosal MNV infection induces lupus in BQ.*Ncf1^{90H}* mice through T cell-dependent germinal center response. a** Timeline of MNV infection by gavage. **b** The quantification of MNV in the small intestine, colon, and spleen on day 7 by RT-qPCR ($n = 9$ per group). Data were analyzed by the Mann-Whitney test (two-tailed) and presented as mean ± SEM. **c** Titer of anti-MNV antibodies on days 0, 14, 28, 56, 70, and 84 in females (MNV-R90: $n = 6$; MNV-90H: $n = 13$) and males (MNV-R90: $n = 5$; MNV-90H: n = 6). Data were analyzed by the Mann-Whitney test (two-tailed) and presented as mean ± SEM. **d** IgG production in femals (R90: $n = 6$; 90H: $n = 13$; MNV-R90: $n = 6$; MNV-90H: $n = 13$) and males (R90: $n = 5$; 90H: $n = 6$; MNV-R90: $n = 5$; MNV-90H: $n = 6$). Data were analyzed using one-way ANOVA and presented as mean ± SEM. **e** The correlation between IgG and anti-MNV antibodies in wild-type *Ncf1^{R90}* mice ($n = 11$) and *Ncf1^{90H}* mice ($n = 12$). Data were analyzed using the Pearson correlation test. **f** Anti-ssRNA antibodies on day 56 in females (R90: $n = 5$; 90H: $n = 5$; MNV-R90: $n = 5$; MNV-90H: $n = 13$) and males (R90: $n = 5$; 90H: $n = 5$; MNV-R90: $n = 5$; MNV-90H: $n = 13$). Data were analyzed using one-way ANOVA and presented as mean ± SEM. **g** Anti-dsDNA antibodies on day 56 in females (R90: $n = 5$; 90H: $n = 5$; MNV-R90: $n = 5$; MNV-90H: $n = 13$) and males (R90: $n = 5$; MNV-R90: $n = 5$; MNV-90H: $n = 6$). Data were analyzed using one-way ANOVA and

presented as mean ± SEM. **h** Anti-Sm/RNP antibodies on day 56 in females (R90: $n = 5$; 90H: $n = 5$; MNV-R90: $n = 5$; MNV-90H: $n = 13$) and males (R90: $n = 5$; 90H: $n = 5$; MNV-R90: $n = 5$; MNV-90H: $n = 6$). Data were analyzed using one-way ANOVA and presented as mean ± SEM. **i** Proteinuria on day 56 in females (R90: $n = 6$; 90H: $n = 13$; MNV-R90: $n = 6$; MNV-90H: $n = 13$) and males (R90: $n = 5$; 90H: $n = 6$; MNV-R90: $n = 5$; MNV-90H: $n = 6$). Data were analyzed using one-way ANOVA and presented as mean ± SEM. **j** The ratio of Tcm or Tvm cells in CD4^+ Foxp3^+ in the PPs (MNV-R90: $n = 11$; MNV-90H: $n = 19$). Data were analyzed by the Mann-Whitney test (two-tailed) and presented as mean ± SEM. **k, l** The ratio of Tfh cells and Tfr cells in CD4^+ FOXP3^+ cells (MNV-R90: $n = 11$; MNV-90H: $n = 19$). Data were analyzed by the Mann-Whitney test (two-tailed) and presented as mean ± SEM. **m** The ratio of Trm cells in CD4^+ CD44^{hi} cells (MNV-R90: $n = 11$; MNV-90H: $n = 18$). Data were analyzed by the Mann-Whitney test (two-tailed) and presented as mean ± SEM. **n** The ratio of GC-B cells in IgD^{lo} B cells (MNV-R90: $n = 11$; MNV-90H: $n = 19$). Data were analyzed by the Mann-Whitney test (two-tailed) and presented as mean ± SEM. **o** The ratio of LLPCs in IgD^{lo} CD138^+ Sca-1^+ cells (MNV-R90: $n = 11$; MNV-90H: $n = 19$). Data were analyzed by the Mann-Whitney test (two-tailed) and presented as mean ± SEM.

kept and bred in the specific pathogen-free facility of Laboratory Animal Center of Southern Medical University and Comparative Medicine Annex (KM-A) of Karolinska Institutet except the infection experiments which were operated in the barrier of Comparative Medicine's Annex (KM-F) of Karolinska Institutet. The facilities have a climate-controlled environment with a 14 h light/10 hrs dark cycle. The animals were housed in individually ventilated polystyrene cages containing enrichments with standard chow and water given *ad libitum*. For all the experiments, 4- to 8-week-old age- and sex-matched mutated mice and wild-type littermate controls were used. Experimental mice were killed by cervical dislocation. All the experimental procedures were approved by the Southern Medical University and Karolinska Institutet Animal Ethics Board (Guangzhou, China, permit number: L2020013 or Stockholm, Sweden, permit number: Dnr 23517-2022, 10523-2022 and 2660-2019). All animal experiments were performed according to the ARRIVE guidelines[50].

### Murine norovirus preparation

The MNV strain Berlin/06/06DE S99 was cultured in the permissive macrophage mouse cell line RAW 264.7 grown in Dulbecco's minimum essential medium (DMEM, no pyruvate; catalog no. FG 0435, Biochrom), supplemented with 10% low endotoxin fetal bovine serum (Hyclone FBS, SH30088), 1% non-essential amino acids (Life Technologies, 11140-050) and 5% penicillin/streptomycin (Life Technologies, 15140122). Overnight cell cultures in T75 flasks were infected with MNV and incubated for 24 h, after which the supernatant was harvested, and cells were removed by centrifugation. Virus stock was stored at −80 °C. Viral preparations were derived from cell cultures. Viral infectivity was evaluated by a cell infectivity assay, determining the 50% tissue culture infective dose (TCID50) using 24-well plates and negative sense RNA detection for confirmation of infection[51].

### Establishment of lupus and arthritis models

To develop the pristane-induced lupus model, mice were injected intraperitoneally with a single dose of 500 μL pristane (MilliporeSigma, P2870) and followed for 6 months. For environmental MNV infections, bedding from the MNV-positive cage was added weekly in equal amounts as clean autoclaved bedding to the cages that housed 8-week-old MNV-free *Ncf1^{90H}* and *Ncf1^{R90}* mice obtained from the SPF animal facility[52]. MNV RNA was isolated from the feces and sequenced (Table S2, 3). For the mucosal MNV-induced model, all mice were infected by the MNV-S99 strain via oral gavage inoculation at a dose of $2 \times 10^5$ TCID$_{50}$ per mouse. For the non-mucosal MNV-induced model, the mice were injected with $3 \times 10^5$ TCID$_{50}$ by both intravenous and intraperitoneal injections, respectively.

### Evaluation of arthritis and lupus

Arthritis was monitored using a macroscopic scoring system in which 1–5 points were given[7]. Briefly, 5 points were given to each visibly inflamed (erythema and swelling) ankle or wrist and 1 point to each inflamed toe. Histopathological evaluation on the ankle joints and kidneys collected at the endpoint by H&E staining was also performed. For assessing the glomerular deposits, PBS-flushed kidneys were embedded in OTC Tissue-Tek (Sakura, 4583) compound, snap frozen, and stored at −80°C. Five-micrometer cryo-sections were cut, fixed, permeabilized with acetone, and stained with Alexa Fluor 488-conjugated anti-mouse complement component C3 (1:200 dilutions; Cedarlane, CL7503AF4) or Alexa Fluor 488-conjugated goat anti−mouse IgG specific for Fcγ fragment (1:500 dilutions, Jackson ImmunoResearch, 115-545-071). Glomerular deposits were recorded by LSM880 with Airyscan, and evaluated by two blinded individual scorers using immunofluorescence microscopy Zen Blue v3.1 and a semi-quantitative scoring system. Proteinuria was determined with semiquantitative urine testing strips (Uristix, 2857) using midstream urine.

### ELISA and ELISpot

For DNA, RNA and Sm/RNP ELISA, plates were coated with 1 μg mL$^{-1}$ Sm/RNP (Avivasysbin, OPMA04153-1000 IU) or 20 μg mL$^{-1}$ poly-L-lysine (MilliporeSigma, P2658) before the addition of 20 μg mL$^{-1}$ of calf thymus DNA (Sigma-Aldrich, D7290), 25 μg mL$^{-1}$ RNA (Thermo Scientific, AM7120G). Plates were then blocked in PBS/2% FBS (Thermo Fisher Scientific, 26140079) for 2 h at room temperature. Sera were 1:50 diluted in PBS/2% FBS. Bound IgG was detected with HRP-conjugated goat anti-mouse IgG (H + L) (1:4,000 dilutions; Southern Biotech, 1031-05) followed by addition of substrate solution (Seramun Diagnostica, S-100-TMB). Samples were acquired with Biotek Gen5 v1. 05.

For anti-COL2 antibody or IgG detection, plates were coated with 10 μg mL$^{-1}$ of purified COL2, goat anti-mouse IgG, or human ads-UNLB (SouthernBiotech, 1030-01) in PBS at 4 °C overnight. After blocking with 5% BSA in PBS at room temperature (RT) for 2 h, diluted polyclonal antibodies purified from QD mouse antibodies or 1:1000 diluted serum samples were incubated at RT for 2 h. The antibody titers were evaluated using HRP-conjugated goat anti-mouse-IgG, -IgG2b, and -IgM (1:4,000 dilutions; Southern Biotech, 1030-05, 1090-05, 1020-05). Samples were acquired with Biotek Gen5 v1. 05.

Detection of anti-MNV antibodies in the sera was performed using an ELISA Kit (Express Biotech, MD, 595-631C) or plates coated with 10$^4$ copies/100 μL of MNV-S99 solution, after blocking with 3% BSA at 37 °C for 2 h. Sera was diluted 1:1000 in PBS/2% FBS. Bound IgG was detected with goat anti-mouse IgG, human ads-BIOT (1:4000 dilutions;

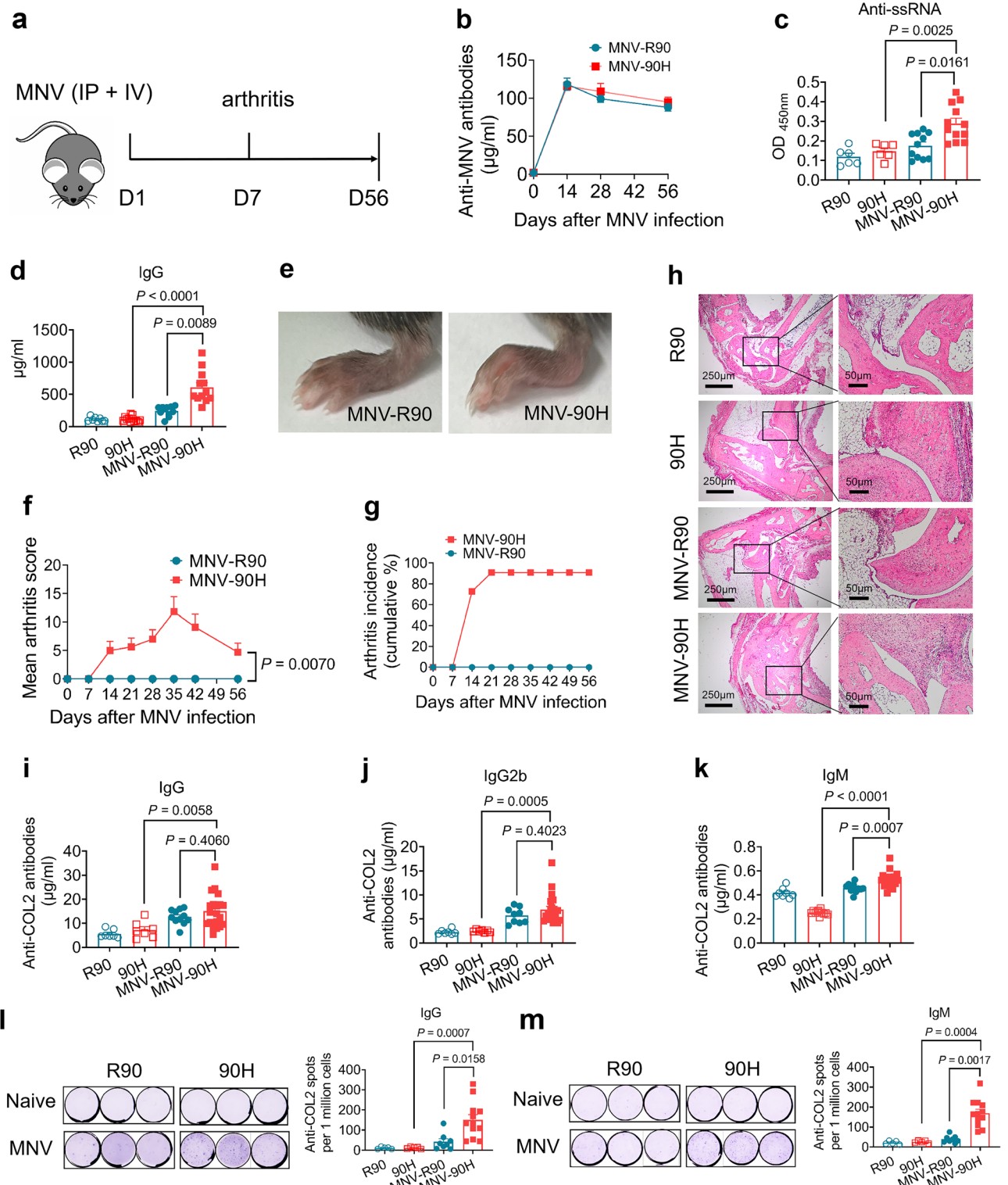

**Fig. 5 | Non-mucosal MNV infection induces lupus arthritis in BQ. *Ncf1^{90H}* mice.**
**a** Timeline of MNV infection in male mice by i.v. and i.p. injection. **b** Anti-MNV antibodies on days 0, 14, 28, and 56. **c** Anti-ssRNA antibodies on day 21 (R90: $n = 6$; 90H: $n = 6$; MNV-R90: $n = 11$; MNV-90H: $n = 12$). Data were analyzed using one-way ANOVA and presented as mean ± SEM. **d** IgG production on day 21 (R90: $n = 8$; 90H: $n = 12$; MNV-R90: $n = 11$; MNV-90H: n = 12). Data were analyzed using one-way ANOVA and presented as mean ± SEM. **e** Representative paws on day 21. **f** Mean arthritis score (MNV-R90: $n = 8$; MNV-90H: $n = 11$). Data were analyzed by the Mann-Whitney test (two-tailed) and presented as mean ± SEM. **g** Arthritis incidence ($n = 9$ per group). **h** Representative HE staining of the joints ($n = 4$, magnification x 2.5 and x 10). **i** Anti-collagen II (COL2) IgG antibodies (R90: $n = 7$; 90H: $n = 7$; MNV-R90:

$n = 11$; MNV-90H: $n = 20$). Data were analyzed using one-way ANOVA and presented as mean ± SEM. **j** Anti-COL2 IgG2b antibodies (R90: $n = 7$; 90H: $n = 7$; MNV-R90: $n = 9$; MNV-90H: $n = 20$). Data were analyzed using one-way ANOVA and presented as mean ± SEM. **k** Anti-COL2 IgM antibodies (R90: $n = 8$; 90H: $n = 11$; MNV-R90: $n = 11$; MNV-90H: $n = 18$). Data were analyzed using one-way ANOVA and presented as mean ± SEM. The level of anti-COL2 antibodies was measured in mouse sera on day 21 by ELISA. **l, m** Representative B cell-ELISpot and statistics of total anti-COL2 IgG and IgM ASCs in the spleen were shown (R90: $n = 5$; 90H: $n = 5$; MNV-R90: $n = 8$; MNV-90H: $n = 12$). Data were analyzed using one-way ANOVA and presented as mean ± SEM.

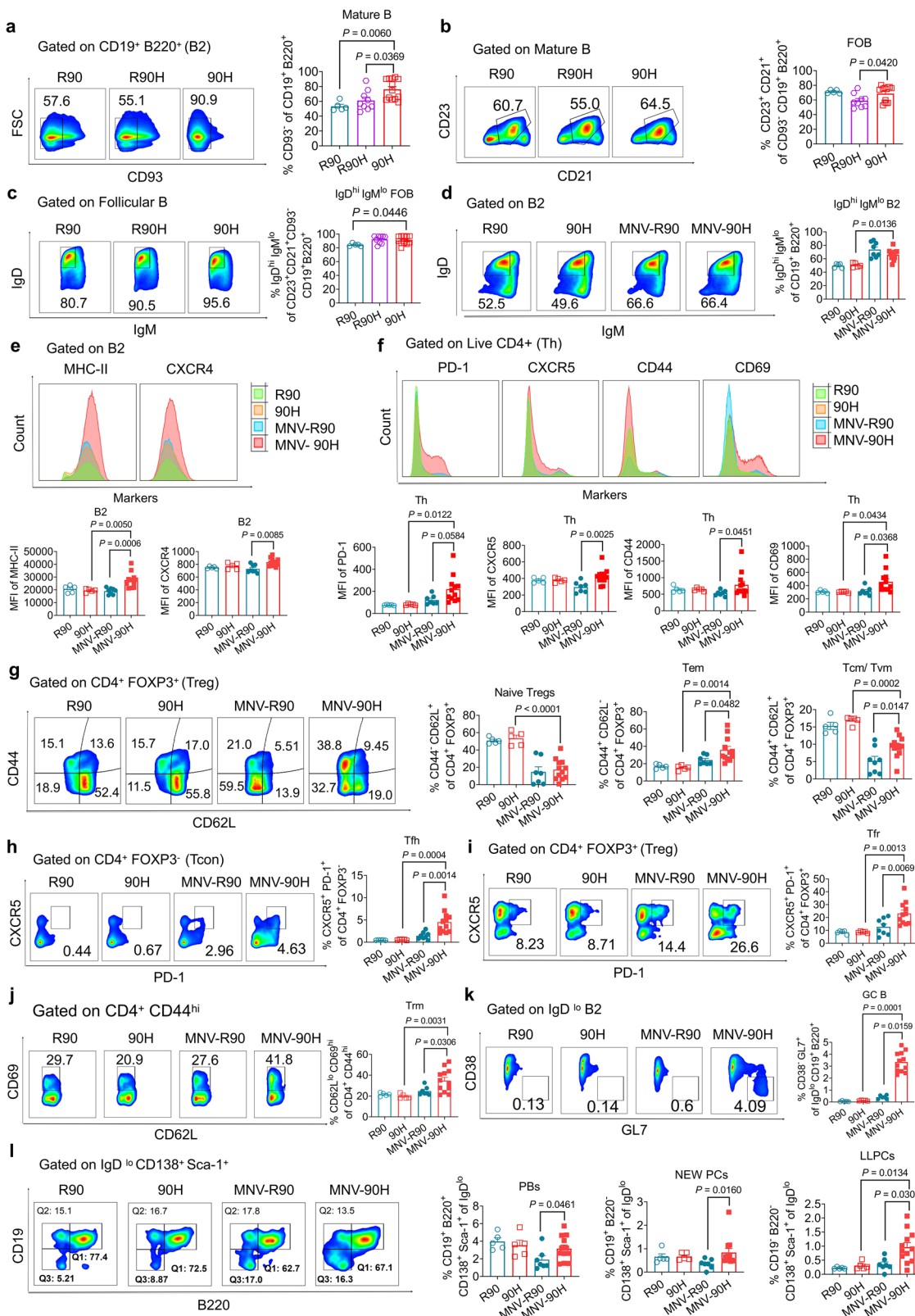

Southern Biotech, 1030-08), followed by Streptavidin-HRP (1:800 dilutions, Southern Biotech, 7105-05) and substrate solution (Seramun Diagnostica, S-100-TMB)[53]. Samples were acquired with Biotek Gen5 v1. 05.

For ELISpot, plates were activated with 15 µL of 35% ethanol, washed two times with PBS, and coated with 10 µg mL⁻¹ COL2 in PBS at 4 °C overnight. After washing, $1 \times 10^6$ splenocytes were plated per well

in complete RPMI media containing 10% FCS and penicillin/streptomycin and incubated for 6 h at 37°C. After washing, biotinylated goat anti-mouse IgG (1:1000 dilutions, Southern Biotech, 1030-08) or IgG2b (1:1000 dilutions, Southern Biotech, 1090-08) was added to the wells and incubated at RT for 2 h. ExtrAvidin conjugated alkaline phosphatase (Sigma-Aldrich, E2636) diluted in PBS was added and incubated for 30–45 min. 100 µL of substrate BCIP/Nitroblue Tetrazolium

**Fig. 6 | The *Ncf1[90H]* allele promotes maturation and differentiation of B and T cells. a** The frequency of B and T cells in the spleen of naïve mice. The ratio of mature B cells (CD93⁻, gated on CD19⁺ B220⁺) in B2 cells (CD19⁺ B220⁺) (R90: *n* = 5; R90H: *n* = 9; 90H: *n* = 13). Data were analyzed using one-way ANOVA and presented as mean ± SEM. **b** The ratio of follicular B cells (CD23⁺ CD21⁺, gated on CD19⁺ B220⁺ CD93⁻) in B2 cells (R90: *n* = 5; R90H: *n* = 9; 90H: *n* = 13). Data were analyzed using one-way ANOVA and presented as mean ± SEM. **c** The ratio of IgD^hi IgM^lo follicular B cells (IgD^hi IgM^lo, gated on CD23⁺ CD21⁺ CD19⁺ B220⁺ CD93⁻) in B2 cells (R90: *n* = 5; R90H: *n* = 9; 90H: *n* = 13). Data were analyzed using one-way ANOVA and presented as mean ± SEM. **d** Mice were infected with MNV for 21 days by i.v. and i.p. injection. The frequency of IgD^hi IgM^lo from B2 cells (R90: *n* = 5; 90H: *n* = 5; MNV-R90: *n* = 8; MNV-90H: *n* = 12). Data were analyzed using one-way ANOVA and presented as mean ± SEM. **e** The histogram and MFI value of MHCII, and CXCR4 on B2 cells (R90: *n* = 5; 90H: *n* = 5; MNV-R90: *n* = 8; MNV-90H: *n* = 12). Data were analyzed using one-way ANOVA and presented as mean ± SEM. **f** The expression of PD-1, CXCR5, CD44, CD69 and CD62L on Th cells (CD4⁺) (R90: *n* = 5; 90H: *n* = 5; MNV-R90: *n* = 8; MNV-90H: *n* = 12). Data were analyzed using one-way ANOVA and presented as mean ±

SEM. **g** Naïve Tregs (CD4⁺ FOXP3⁺), Tem cells (CD62L⁻ CD44⁺, gated on CD4⁺ FOXP3⁺), Tcm/Tvm cells (CD62L⁺ CD44⁺, gated on CD4⁺ FOXP3⁺) (R90: *n* = 5; 90H: *n* = 5; MNV-R90: *n* = 8; MNV-90H: *n* = 12). Data were analyzed using one-way ANOVA and presented as mean ± SEM. **h** Tfh (CXCR5⁺ PD-1⁺, gated on CD4⁺ FOXP3⁻) (R90: *n* = 5; 90H: *n* = 5; MNV-R90: *n* = 8; MNV-90H: *n* = 12). Data were analyzed using one-way ANOVA and presented as mean ± SEM. **i** Tfr cells (CXCR5⁺ PD-1⁺, gated on CD4⁺ FOXP3⁺) (R90: *n* = 5; 90H: *n* = 5; MNV-R90: *n* = 8; MNV-90H: *n* = 12). Data were analyzed using one-way ANOVA and presented as mean ± SEM. **j** Trm cells (CD62L⁻ CD69^hi, gated on CD4⁺ CD44^hi) (R90: *n* = 5; 90H: *n* = 5; MNV-R90: *n* = 8; MNV-90H: *n* = 12). Data were analyzed using one-way ANOVA and presented as mean ± SEM. **k** GC-B cells (CD38^lo GL7⁺ cells, gated on IgD⁻ B220⁺ CD19⁺ population) (R90: *n* = 5; 90H: *n* = 5; MNV-R90: *n* = 8; MNV-90H: *n* = 12). Data were analyzed using one-way ANOVA and presented as mean ± SEM. **l** ASCs (CD138⁺ Sca-1⁺ IgD⁻) subpopulation: PBs (CD19⁺ B220⁺), NEW PCs (CD19⁺ B220⁻), and LLPCs (CD19⁻ B220⁻) (R90: *n* = 5; 90H: *n* = 5; MNV-R90: *n* = 8; MNV-90H: *n* = 12). Data were analyzed using one-way ANOVA and presented as mean ± SEM.

(Sigma-Aldrich, B3804) per well (1 tablet in 10 mL ddH₂O and pre-filtered with 0.45 μm) was added and incubated in the dark for 10–15 min. When visible spots appeared, the wells were rinsed thoroughly with tap water. The plastic bottom from the plate was removed, rinsed further, and dried in darkness. Scanned wells (ImmunoScan) were recorded by BioSpot Software, and analyzed with ImmunoSpot software (Cellular Technology).

## Flow cytometry

Flow cytometry analyzes were performed[54]. Briefly, organs were collected, mashed, and filtered through 45 μM filter to obtain single-cell suspensions in PBS. To prepare single-cell suspensions, the perfused kidneys were digested with 1 mg/mL collagenase (Roche, 11088866001) and 0.1 mg/mL DNase I (Roche, 10104159001) in a 37 °C water bath for 45 min. Red blood cells were lysed using ammonium-chloride-potassium (ACK) buffer (homemade), cells were counted on a Sysmex, and purified rat anti-mouse Fc-block (CD16/CD32, 24G2, homemade, BD Biosciences; ≤1 μg/million cells in 100 μl) was added for 10 min at RT. Surface antigens were stained with fluorescently labeled antibodies. All the antibodies were used with 0.2 μg per million cells in 100 μl volume. To stain immune cells in the peritoneal cavity, spleens, and kidneys, CD45-HV500, Ly6C-BV605, Ly6G-APC, F4/80-PE, B220-Fluor50, CD19-PE, CD3-APC, CD4-PerCP/Cyanine5.5 and CD8a-FITC antibodies (Biolegend) were used. For the staining B and T cells, CD19-PE-Cy7, B220-PB, CD93-PE, CD21-APC, CD23-PerCP-Cy5.5, IgM-BV605, IgD-BV650, B220-APC, MHCII-FITC, CXCR4-PerCP-Cy5.5, CXCR5-BV421, CD4-BV605, CD44-AF700, CD62L-FITC, CXCR5-BV421, PD-1-PE-Cy7, FOXP3-APC, CD93-PE, CD19-AF700, CD138-BV605, GL7-APC, CD38-PE, Sca-1-PE-Cy7, CD69-APC, CXCR5-PB, PD-1-PE, CCR6-PE-Cy7, CXCR3 (CD183)-PerCP-Cy5.5, CD69-PE antibodies (Biolegend) were used. For detection of STAT1/p-STAT, peritoneal exudates cells were collected one day after MNV intraperitoneal injection at the dose of 3 × 10⁵ TCID₅₀, stained with CD45-HV500, CD11b-PE-Cy7, Ly6C-BV605, Ly6G-APC, F4/80-FITC (Biolegend), washed and fixed with Cytofix buffer (BD Biosciences) for 10 min at 37 °C, permeabilized with Phosflow Perm Buffer III on ice for 30 min, washed twice and then stained with PE Mouse anti-Total Stat1 (1:50 dilutions; BD Phosflow, N-Terminus, 558537) and BV421 Mouse anti-p-STAT1^Tyr701 antibody (1:50 dilutions; BD Phosflow, 566238). For the detection of TLR7 and 9, splenocytes were collected, CD45-PerCP-Cy5.5, CD11b-PB, F4/80-APC, Ly6C-BV605, CD11c-PE-Cy7, CD19-AF700, B220-PB, PDCA1 (CD317)-APC antibodies (Biolegend) were used. Cells were fixed, permeabilized, and stained with TLR7-PE (BioLegend, clone: A94B10, 160003), and TLR9-FITC (BioLegend, clone: S18025A, 159107) antibodies. Dead cells were excluded using FVS780 (BD Biosciences, 565388) or a fixable near-IR dead cell stain kit (Thermo Fisher Scientific, L10119). Samples were either acquired with an Attune v5.2.0 flow cytometer (Thermo

Fisher) or using LSR Fortessa (BD Biosciences). Samples were analyzed with FlowJo version 10.6.

## Bone marrow-derived macrophages (BMDMs)

Mouse BMDMs were obtained from the differentiation of monocytes recovered from the femur and tibia of 6 to 8- week- old male B6N.Q.*Ncf1[R90]*, B6N.Q. *Ncf1[90H]*, B6N.Q and B6N.Q.*Ncf1[m1J]* mice. Bone marrow was flushed from the cavity, and single-cell suspensions were generated and cultured in RPMI-1640 medium (Gibco, 21875091) with M-CSF (20 ng/mL, Peprotech, 315-02-10) and 10% fetal bovine serum for obtaining macrophages. The mature macrophages (M0) were confirmed by identifying the makers (F4/80⁺ CD11b⁺) using flow cytometry. For ROS measurement, BMDMs were cultured, collected, washed with PBS and used. To measure the intracellular ROS production, a DCFH-DA fluorescent probe kit (Byotime, S0033S) was used with or without Rosup stimulation. For extracellular ROS measurement, BMDMs were treated with 100 μL of isoluminol reagent buffer, as previously described[10]. Data was collected using a Biotek plate reader.

## Immunofluorescence

BMDMs from the naïve and IFN-α stimulated mice were cultured in confocal dishes, fixed with 4% paraformaldehyde, blocked with 10% FCS/PBS buffer, and then incubated with the primary antibodies: mouse anti-p47phox (D-10) antibodies (Santa Cruz Biotechnology, sc-17845), or STAT1 (CST, clone: D1K9Y, 65748), p-STAT1^Tyr701 (CST, clone: 58D6, 88845), STAT3 (CST, clone: 79D7, 4904), and p-STAT3^Tyr705 (CST, clone: Tyr705, 9131) specific rabbit antibodies (1:1000 dilutions). Subsequently, cells were incubated with either goat anti-mouse IgG (H + L) (1:1000 dilutions; CST, 4408) or goat anti-rabbit IgG (H + L)-Alexa Fluor 488 (1:1000 dilutions; CST, 4412), followed by counterstaining with 4′, 6-diamidino-2-phenylindole DAPI (Vector, CA) and dried for 30 min before scanning under a confocal microscope (LSM880 with Airyscan, CarlZeiss, Germany).

## Western blot

For experiments in vitro, 1000 U/mL of IFN-α (BioLegend, 752804) was used to stimulate BMDMs (M0 cells) for 30 min, 1000 U mL⁻¹ of IFN-α and 25 μM of GSK2795039 (MCE, HY-18950) or 1000U mL⁻¹ of IFN-α and 10 μM of H₂O₂ was used. Total protein lysates from cells or tissues were extracted using the radioimmunoprecipitationassay (RIPA) solution with a cocktail of protease and phosphatase inhibitors (Roche, 04693116001, 04906837001). The final protein concentration of each sample was determined with a BCA kit (ThermoFisher Scientific, 23225). The supernatants from protein lysates were subjected to SDS-polyacrylamide gel electrophoresis (PAGE) (Invitrogen, NP0321BOX). The following primary antibodies were used: mouse ani-NCF1

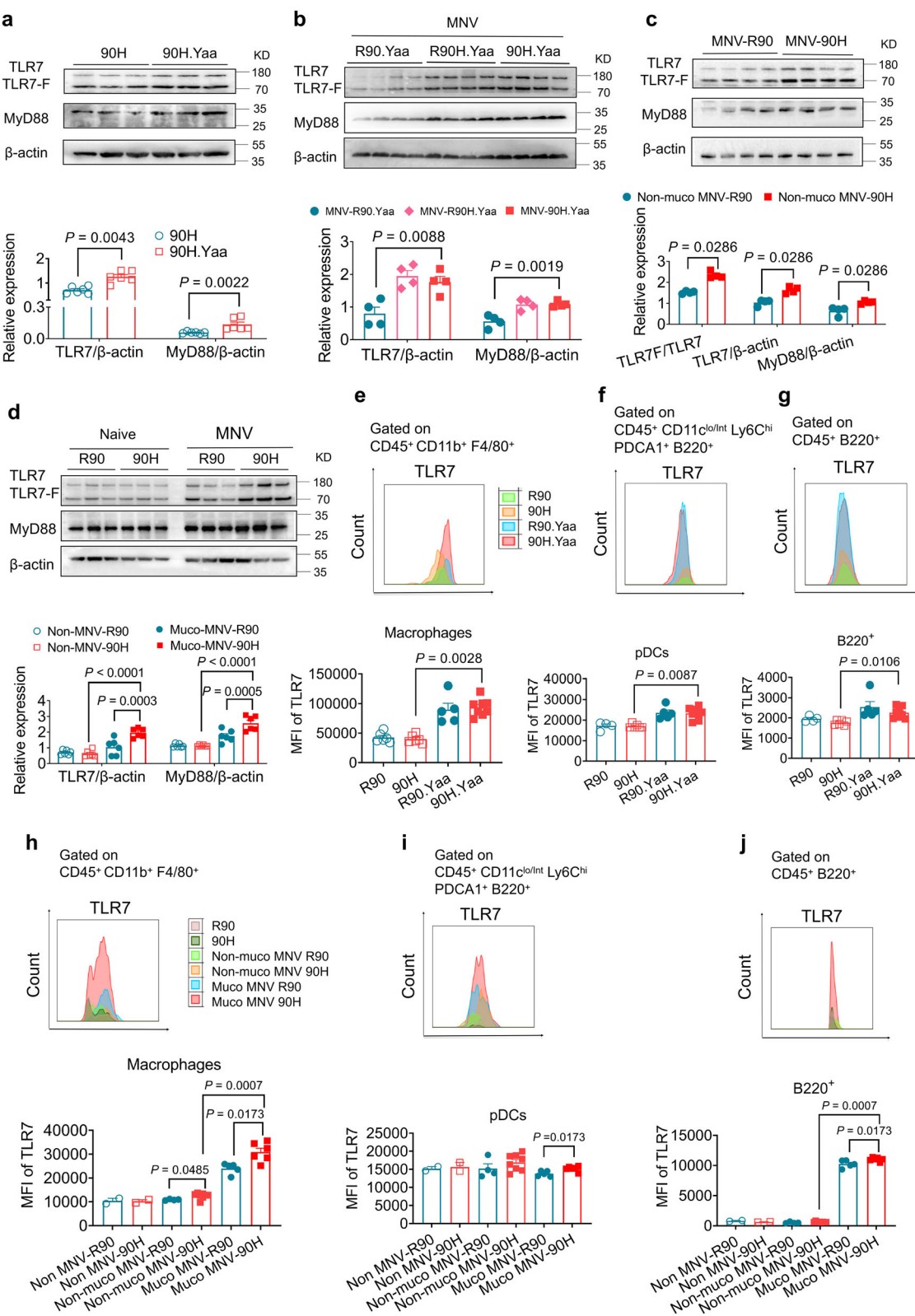

(p47phox) (D-10) antibodies (Santa Cruz Biotechnology, sc-17845) or JAK1 (CST, 3332), p-JAK1 (CST, 3331), JAK2 (CST, 3230), p-JAK2 (CST, 3771), STAT1 (CST, clone: D1K9Y, 65748), p-STAT1$^{Tyr701}$ (CST, clone: 58D6, 88845), STAT3 (CST, clone: 79D7, 4904), and p-STAT3$^{Tyr705}$ (CST, clone: Tyr705, 9131) specific rabbit antibodies (1:1000 dilutions), TLR-7 (CST, D7, 5632) and MyD88 (CST, D80F5, 4283) specific rabbit antibodies, and β-Actin rabbit antibody (CST, 4967) at 4°C overnight. The signal was detected using the goat anti-mouse IgG conjugated with HRP (1:4000 dilutions; Southern Biotech, 1031-05) or anti-rabbit IgG conjugated with HRP (1:4000 dilutions; Southern Biotech, 4030-05) secondary antibody. The data were collected by Bio-Rad CFX Manager v3.1 and analyzed by Image J.

**Fig. 7 | The NCF1$^{90H}$ allele upregulates MNV-induced TLR7. a** Immunoblot analysis of intact TLR7 protein and MyD88 protein in the spleen of naïve male 90H mice with or without *Yaa* locus ($n = 6$ per group). Data were analyzed by the Mann-Whitney test (two-tailed) and presented as mean ± SEM. **b** Immunoblot analysis of TLR7 and MyD88 protein in environmentally MNV-infected 90H mice with *Yaa* locus on day 70 ($n = 4$ per group). Data were analyzed by the Mann-Whitney test (two-tailed) and presented as mean ± SEM. **c** Immunoblot analysis of intact TLR7 protein, cleaved TLR7 fragment (TLR7-F), and MyD88 in non-mucosal MNV-infected 90H mice ($n = 4$ per group). Data were analyzed by the Mann-Whitney test (two-tailed) and presented as mean ± SEM. **d** Immunoblot analysis of intact TLR7 and MyD88 proteins in non-MNV and mucosal MNV infected mice ($n = 6$ per group). Data were analyzed using one-way ANOVA and presented as mean ± SEM. **e–g** Flow cytometric analysis of TLR7 in the spleen of naïve mice with and without *Yaa* locus in macrophages (CD45$^+$ CD11b$^+$ F4/80$^+$) (R90: $n = 6$; 90H: $n = 6$; R90.Yaa: $n = 5$;

90H.Yaa: $n = 8$), pDCs (CD45$^+$ CD11c$^{lo/int}$ Ly6C$^{hi}$ PDCA-1$^+$ B220$^+$) (R90: $n = 4$; 90H: $n = 6$; R90.Yaa: $n = 5$; 90H.Yaa: $n = 8$), and CD45$^+$B220$^+$ splenocytes (R90: $n = 4$; 90H: $n = 6$; R90.Yaa: $n = 5$; 90H.Yaa: $n = 8$). Data were analyzed by the Mann-Whitney test (two-tailed) and presented as mean ± SEM. **h–j** Flow cytometric analysis of TLR7 in macrophages (Non MNV-R90: $n = 2$; Non MNV-90H: $n = 2$; Non-muco MNV-R90: $n = 4$; Non-muco MNV-90H: $n = 8$; Muco MNV-R90: $n = 5$; Muco MNV-90H: $n = 6$), pDCs (Non MNV-R90: $n = 2$; Non MNV-90H: $n = 2$; Non-muco MNV-R90: $n = 4$; Non-muco MNV-90H: $n = 8$; Muco MNV-R90: $n = 5$; Muco MNV-90H: $n = 6$), and B220$^+$ cells (Non MNV-R90: $n = 2$; Non MNV-90H: $n = 2$; Non-muco MNV-R90: $n = 4$; Non-muco MNV-90H: $n = 8$; Muco MNV-R90: $n = 5$; Muco MNV-90H: $n = 6$) from the spleen of mice without MNV infection, with non-mucosal MNV infection, and mucosal MNV infection. Data were analyzed using one-way ANOVA and presented as mean ± SEM.

## Quantitative real-time PCR

Spleen, kidney, and bone marrow cells were obtained from age- and sex-matched mutated and wild-type mice after carbon dioxide asphyxiation. Bone marrow macrophages were harvested and cultured as described above. Total RNA was isolated by using TRIzol Reagent (Invitrogen, 15596026CN), and complementary DNA (cDNA) was synthesized using the First Strand cDNA Synthesis Kit (ThermoFisher Scientific, K1622). RT-qPCR was performed by Agilent Strata gene Mx3005P with FastStart Universal SYBR Green Master (Roche, 4913914001) to assess gene expression. The data were analyzed with Microsoft Office 16. The relative gene expression normalized by β-actin was calculated using the $2^{-\triangle\triangle CT}$ method. The primers used are described in Supplementary Table 3 and 4.

The small intestine, colon, and spleen were prepared for RNA extraction[52]. Detection of MNV was done with an MNV-specific primer pair targeting a 396 bp region at the 5'end of MNV ORF2 (VP1) using an RT-PCR kit (Thermo Fisher Scientific)[55]. E.Z.N.A.R Viral RNA Kit (Omega, R6874-01) was used to extract the viral RNAs from the feces of MNV-infected mice. The SuperScript® First-Strand Synthesis System for RT-PCR (Invitrogen, 11904-018) was used to synthesize first-strand cDNA. Different regions of the MNV genomes were amplified by applying 12 pairs of oligonucleotide primers designed based on the sequence from the MNV BJ 10-2062 strain (Table S3). After purification and cloning into the pMD18-T vector (TaKaRa, Japan), the PCR products were sequenced by Beijing Sunbiotech Co., Ltd. (Beijing, China). Lasergene software (DNAstar Inc., USA) was used to assemble and analyze the sequence data. Clustal X 2.1 and MEGA 6 programs were used to perform multiple sequence alignments and phylogenetic analyzes, respectively. The genome sequence of the isolated strain was registered in GenBank (Accession Number: MT358379). Strand-specific real-time RT-qPCR assay for the detection and quantitation of murine norovirus RNA as described previously[56].

## Statistical analysis

Quantitative data were expressed as mean ± SEM. The statistical analysis of differences between the experimental groups was performed using a one-way analysis of variance (ANOVA) or Student's t-test. ROS assays were calculated using Mann-Whitney or Kruskal-Wallis tests. Clinical arthritis and histology scores were analyzed using the Mann-Whitney U-test, whereas cellular analyzes were analyzed with the One-Way ANOVA combined with the Tukey post-test. Pearson's correlation test was used for analyzing the correlation of antibody responses using GraphPad Prism Software Version 9.5.0 (GraphPad Software, Inc.). All data shown are mean ± SEM. The p values less than 0.05 were considered significant.

## Reporting summary

Further information on research design is available in the Nature Portfolio Reporting Summary linked to this article.

## Data availability

All data are included in the Supplementary Information or available from the authors, as are unique reagents used in this Article. The raw numbers for charts and graphs are available in the Source Data file whenever possible. Source data are provided with this paper.

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

## Acknowledgements

This study was supported by the KI Foundation for Virus Research (2023-00122, YL), KI Foundation funds for rheumatology research (2023-02710, YL), Science and Technology Major Project District-School Cooperation Outstanding Youth Fund (Shenzhen Nanshan District Health System (NSZD) (NSZD2023062, ZL), the EU COSMIC Marie Curie grant (765158, RH), the Swedish Research Council (2023-06482, RH), Southern Medical University (SMU) grant (C1034211, RH), the Natural Science Foundation of China (No.32070913, 82471830, W2431021, RH), Vetenskapsrådet (VR) (2024-02575, RH), NovoNordisk (NNF24OC0090035, RH), Leo Foundation (LF-OC-22-001023, RH), Cancer foundation (222350Pj01H, RH), and KAW (2019.0059, RH). We thank prof. Patrik Medstrand for collaboration of MNV strain Berlin/06/06DE S99 production.

## Author contributions

Y.L. and R.H. designed the experiments. Y.L. established models and did most of the experiments. A.C. did the flow cytometry and ELISPOT. Z.L. did histology immunofluorescence staining and western blot. J.L. supervised virus production. M.A. produced and quantified MNV. Q.L., R.X., and H.L. participated in the collection of samples, and the scoring of mice. H.L. helped with the collection of samples and the detection of proteinuria. D.L. and J.X. participated in breeding mice. KSN contributed to mice breeding and mice injection. L.M. was involved in the genotyping of the mice. Y.L. wrote the first draft of the paper with the help of R.H. R.H. supervised the project. All authors approved the study and revised the manuscript.

## Funding

## Competing interests

All authors agree to publish the data in this paper. The authors declare no competing interests. The authors declare an absence of any commercial or financial relationships.
