## [Transparent Peer Review file · Nature Communications]

The systemic lupus erythematosus-associated NCF190H allele synergizes with viral infection to cause murine lupus but also limit virus spread

Corresponding Author: Professor Rikard Holmdahl

Version 1:

Reviewer comments:

Reviewer #1

(Remarks to the Author)

In their current manuscript, Li et al. investigate the significance of a point mutation (R90H) in neutrophil cytosolic factor 1 (NCF1) in the pathogenesis of systemic lupus erythematosus (SLE). This mutation has been linked to in humans, and knock-in of a H90 gene variant increases the propensity of mice to develop a lupus-like phenotype. Li et al. have now made the serendipitous observation that infection of 2-year-old Ncf1R90H mice with murine norovirus (MNV) promotes a lupus-like phenotype. They provide a detailed characterization of this phenotype with arthritis, proteinuria, an SLE autoantibody signature, all of which are neither present in wild-type (WT) mice infected with MNV, nor mutant or wild-type mice not infected with the virus (Fig. 1). Injection of R90H mice with pristane, an agent that is commonly used to induce a lupus-like phenotype in animal models, caused a much more pronounced phenotype in R90H mice than in WT mice regardless of sex (Fig. 2). Compared to WT mice, macrophages of R90H mice produce less intracellular and extracellular reactive oxygen species (ROS) natively and when stimulated experimentally, and their Jak-Stat signaling pathway is (Fig. 3). Fig. 4 explores the consequences of experimental infection of mice with MNV (by oral gavage), and Fig. 5 shows the phenotype of non-mucosal infection of these mice with MNV (by intravenous and intraperitoneal infection). Fig. 6 delineates the germinal center response in R90H mice natively and subsequent to MNV infection, and Fig. 7 demonstrates that TLR7 expression is augmented in R90H mice compared to WT mice, and augmented by MNV infection.

Overall, this is an interesting manuscript which provides additional insights into the pathogenesis of SLE. However, some data that is presented in this manuscript is not new, and it remains unclear how the SLE-facilitating infection of mutant mice with MNV can serve as a general model for the interaction of genetics with environmental factors. Granted, this is an interesting example. However, while providing detailed phenotyping of the MNV-infected Ncf1R90H mice, in part seemingly replicating previously published data from the same group, this work does not include experiments with environmental stimuli other than MNV.

The authors state at the end that their „discovery of MNV as a trigger of lupus in Ncf1R90H mice does not exclude other viruses, environmental factors or genes, to be causative for SLE“. I consider this a limitation, rather than an asset of this work. The conclusion may be true that this „is the first time a disease-causative interaction has been shown between a defined genetic and a specific environmental factor to control a complex autoimmune disease“, but the generalizability of this finding remains to be shown.

Major issues

1. The authors have previously published experiments that are very similar, if not identical to the data shown in Fig. 3 (Li et al., *Antioxidants* 2023;12:148, Pubmed ID 37507888, ref. 36). Please revise the manuscript so that only novel data is shown, and avoid duplication.

2. Notwithstanding the suggested revision of Fig. 3, some of the experiments shown in Fig. 3 need some clarification and better controls. For example, in Fig. 3e, Jak1, Stat1, and Stat3 phosphorylation is normalized to b-actin. Without normalization to the unphosphorylated levels of these proteins, it is possible that the effect shown in Fig. 3e is the same as shown in Fig. 3d. Panel e suggests that the phosphorylation (i.e., activation) response to IFN-alpha is exacerbated in mutant mice, but this cannot be asserted without controlling for unphosphorylated levels. (Fig. 1k, on the other hand, does show normalization to total Jak1 and Stat1 protein levels.)

3. Please clarify what Fig. 3i adds to the main findings of this work. If the authors deem this panel necessary, they should please consider including non-IFN-alpha as well as wildtype control conditions.

4. Please clarify the methodology that was used to obtain the data shown in Fig. 3k.
5. Fig. 4 seems to repeat and expand on Fig. 1 with the major difference that the mice of Fig. 1 were “naturally” infected, while the mice of Fig. 4 were experimentally and deliberately infected. If this is correct, the two figures should please be consolidated.
6. All in all, it becomes clear that R90H mice have a propensity to develop an exaggerated immune activity/immune response, including the serological response to MNV infection. It remains unclear however if this is specific to MNV
7. Could the authors please shed light on how their SPF-housed mice could be accidentally infected with MNV in the first place.

Minor issues

1. The keys to colors and symbols should please be placed in a more visible way in Figs. 1 and 2. Presently the key is included for panel B of Fig.1 only. Fig.2 is lacking a key entirely. Although the colors and symbols are evidently used consistently across figures, each figure should stand on its own.
2. Results section, second sub-heading (“Environmental...”): typo in “arthritis”.

Reviewer #2

(Remarks to the Author)

This paper describes the effect of a missense mutation in the NCF1 gene on the immune response to infection with a murine rotavirus. The NCF1 mutation is associated with the risk of SLE although the mechanism by which this occurs has not been firmly established. In this paper, the authors describe the immunological consequences of a usually non-pathological infection which unexpectedly (and fortuitously for the investigators) was found to promote systemic autoimmunity in the mouse carrying the human SLE risk allele. The experiments are thorough and well-presented and support the conclusions drawn.

Does the missense mutation alter the pattern of tissue expression of NCF1? It is known that interferon can affect NCF1 expression and hence there is scope for interaction between the mutation, viral infection and qualitative and quantitative aspects of expression.

Minor points

Abstract: “We have identified a causal SNP in NCF1” is ambiguous in that the most comprehensive paper describing this association was published by others (ref 9) and the discovery of the causal allele had already taken place and is not part of this paper.

Typo in title in line 201

Explanation of TLR7-F in Figure 7

Reviewer #3

(Remarks to the Author)

Li et al. describe the interesting observation that NCF190H mice develop a lupus-like autoimmune syndrome when infected by murine norovirus, an otherwise not very pathogenic virus. At the same time, the animals clear the virus more quickly, likely due to the exaggerated type 1 IFN response. The paper carefully characterizes the immune response in 90H mice infected either orally (the usual route of infection) or systemically (by IV+IP injection of virus). In both cases, the serum titer of anti-norovirus Abs correlates with the severity of the autoimmunity.

This paper is strictly an observational study. It reports an interesting phenomenon that highlights the interplay between genetic risk alleles for autoimmunity (such as NCF190H) and environmental factors, such as viral infection. However, this group has been studying the NCF190H allele and the related NCF1n1J mice for quite a while, in a variety of autoimmune models in a variety of murine genetic backgrounds. Their mechanistic understanding is that reduced expression of the p47phox protein (encoded by the NCF1 gene) leads to decreased ROS production in myeloid cells. In pDCs, the lower ROS somehow affects signaling through Jak/Stat pathways leading to exaggerated type 1 IFN signaling, setting up an inflammatory loop of more production of type 1 IFNs, with increased signaling in p47phox expressing cells, eventually driving loss of tolerance and development of systemic autoimmunity. This remains a somewhat superficial mechanistic understanding of how the SLE risk allele contributes to autoimmune susceptibility (have the authors ever examined cysteine phosphorylation of specific tyrosine phosphatases such as SHP-1, a known regulator of the Jak/Stat pathway). However, that is not the point of the current paper. It is to simply report the cool finding that infection of these mice with norovirus leads to autoimmune disease.

Some minor improvements may help with this report:

1). All the data is with 90H mice compared to R90 animals. Do the R90H heterozygous mice develop autoimmunity following norovirus infection (either with or without pristane injection)? I am guessing they do not (or the authors would have told us). One would assume that the majority of SLE patients with the NCF1R90H allele are heterozygous (is this true?), so knowing whether the R90H mice have increased autoimmunity following infection would be important. If they do not, the authors need to put a paragraph in the Discussion about why a heterozygous risk allele confers disease susceptibility in humans but not in mice.

2). Where are the viral titers for the oral infection data reported in Fig. 1? They are reported for Fig. 4.

3). Fig. 1 and Fig. 4 both report outcomes with oral infection by norovirus -- Fig. 1 by exposure to infected feces and Fig. 4 by oral gavage. The analysis is a bit different for each figure. Is there any reason to think there would be a difference between these two methods of infection? The data seemed a bit redundant.

4). It wasn't quite clear, but does oral infection by norovirus lead to the anti-COL1 IgG/IgMs reported in the IV+IP model in Fig. 5? It is somewhat difficult to interpret the different immune response analyses reported in the different models. It would seem more logical that the same analysis be completely done on mice infected in different ways.

5). It wasn't quite clear if the pristane + norovirus-infected mice developed a more severe form of autoimmunity than the norovirus infected alone. Fig. 2 only compares pristane against pristane + norovirus. The authors may wish to comment on this.

6) I think Fig. S10A needs a legend. Which are the Yaa-containing mice? The authors don't comment on whether breeding the Yaa allele into the 90H background (as homozygotes) leads to more severe disease than Yaa alone. One would think that it does.

7). For the one sentence about autoimmunity and COVID19 in the Introduction, the authors should check out this very recent report, PMID: 39112696

Overall, this is an interesting phenomenological report. The authors must have been really surprised when, some years ago, their colony of 90H mice suddenly started developing arthritis! They have done a great job figuring out why.

Reviewer #4

(Remarks to the Author)

Li et al demonstrated that infection of MNV with the NCF190H mice, whose SNP causing SLE in humans, induces lupus. The authors also found NCF190H allele upregulated INF- α /JAK1/STAT1 pathway-enhancing type I interferons and antibodies against MNV, and TLR7 in hemopoietic cells being activated by mucosal MNV infection. Taken together, the authors concluded that NCF190H is sensitive to MNV infection, resulting in protection against MNV infection and the development of murine lupus.

This reviewer thinks this manuscript is important to understand the mechanism of SLE development. In particular, identifying MNV as an environmental factor to induces lupus is outstanding, and expect to apply it for the establishment of therapy in humans.

To understand this study deeply, this reviewer expect to add description to answer some questions as below.

1. Although the authors concluded that ssRNA virus infection triggers the development of lupus, the conclusion is a little bit strong to say by only one MNV strain. Have you tested minimal other MNV strains because there is a major distinct pathogenesis, acute and chronic?
2. Does this author know the allele affects functional change in the intestine? Does the expression of MNV receptor proteins change? Although data on infection of MNV by i.g. might support this question, this reviewer would like to know the detailed mechanism in the intestine, which is not fully covered yet in this manuscript.
3. Which region does the sequences in Table 2 show?
4. In Fig S1, the description of p-values isn't correct.
5. The name of the MNV isolate could be unified among Fig .S3 and Table S2.
6. In Fig 1h, the right panel showing IgG and C2 in MNV-90H seems to be too bright rather than the left images. Is this the correct image? On the other hand, the image, Ncf1m1J (? , unreadable due to low resolution) in Fig S7c, seems to be too dark. These images could not support descriptions for each.

Version 2:

Reviewer comments:

Reviewer #1

(Remarks to the Author)

Thank you very much for the amendments made to the manuscript. I still contend that part of this work provides only incremental insights over the Li 2023 paper, but this shall not hinder publication.

Reviewer #2

(Remarks to the Author)

Reviewer #3

(Remarks to the Author)

The authors have made a number of clarifications in the manuscript in response to suggestions from the reviewers. Most of the reviewers felt this was an interesting story. I have no further suggestions.

Reviewer #4

(Remarks to the Author)

This reviewer agrees with the current manuscript including revisions to respond to reviewer comments properly. However, this reviewer would like to suggest a comment related to question 1, which could strengthen this manuscript.

This reviewer agrees with the conclusions which are drawn from data by engaging two MNV strains, Isolate 59591 and Berlin/06/06DE S99. However, these strains are phylogenetically classified comparatively close rather than MNV-1, that is, seem to show similar pathogenicity. Question 1, which this reviewer would like to ask, was if both MNVs, which cause different pathogenesis, acute and chronic infection, result in the same phenotype. This reviewer guesses the answer is NO because the different infection style leads to distinct immune responses. This reviewer accepts the author's claim that testing another MNV strain like the MNV-1 strain is out of aim. On the other hand, this reviewer would like the authors to discuss the possibility of differences in lupus induction between the strains used in this study and MNV-1 to avoid misunderstanding readers that all MNV strains cause the same phenotype.

Response to Reviewer's Comments:

Reviewer #1 (Remarks to the Author)

Background description 1) *In their current manuscript, Li et al. investigate the significance of a point mutation (R90H) in neutrophil cytosolic factor 1 (NCF1) in the pathogenesis of systemic lupus erythematosus (SLE). This mutation has been linked to in humans, and knock-in of a H90 gene variant increases the propensity of mice to develop a lupus-like phenotype. Li et al. have now made the serendipitous observation that infection of 2-year-old*

Authors: Thanks for your valuable comments. Sorry that we didn't make it clear in the paper as, it's a misinterpretation that the mice were 2 years old. The initial observation of induction of lupus of an unknown infection was in Ncf190H mice around 8 weeks old. We have now tried to clarify this with highlights in **Line 166-173**.

Background description 2) *Ncf1R90H mice with murine norovirus (MNV) promotes a lupus-like phenotype. They provide a detailed characterization of this phenotype with arthritis, proteinuria, an SLE autoantibody signature, all of which are neither present in wild-type (WT) mice infected with MNV, nor mutant or wild-type mice not infected with the virus (Fig. 1). Injection of R90H mice with pristane, an agent that is commonly used to induce a lupus-like phenotype in animal models, caused a much more pronounced phenotype in R90H mice than in WT mice regardless of sex (Fig. 2)*

Authors: Thanks for the description. It's however one statement that needs to be better explained. We have now updated the figure and the figure legend to make it more readable. We have not observed that pristane induced severe lupus and arthritis in Ncf1^{90H} housed clean SPF, without MNV infection. In fact, pristane injections did not induce significant/ severe arthritis, or high levels of lupus-associated antibodies, even in Ncf190H mice having BALB/c genes predominantly, in our clean mouse colony. We were surprised and spent several years trying to figure out the reason for this failure. We suddenly found that severe lupus disease developed spontaneously in the colony which we later identified was due to an MNV infection, a disease developing regardless of whether they had been injected with pristane or not. The 90H mice, only infected with MNV at the

beginning, developed more severe and significant diseases compared with WT mice. It demonstrated the development of the disease was completely dependent on the NCF190H allele. This is also what is shown in **Fig 2**, and we have tried to clarify the description in the text and figure legend highlighted (**Line 214-225, and Line 546-553**). In fact, it is that MNV infection enhances pristane-induced lupus and MNV alone (before pristane injection) can also induce slight lupus in the very early stage (**Supplementary Fig. 6d**, also attached in Figures for review) and induce severe lupus in the late stage. We have now tried to clarify this further in the manuscript. Thanks for pointing it out.

Background description 3) *Compared to WT mice, macrophages of R90H mice produce less intracellular and extracellular reactive oxygen species (ROS) natively and when stimulated experimentally, and their Jak-Stat signaling pathway is (Fig. 3). Fig. 4 explores the consequences of experimental infection of mice with MNV (by oral gavage), and Fig. 5 shows the phenotype of non-mucosal infection of these mice with MNV (by intravenous and intraperitoneal infection). Fig. 6 delineates the germinal center response in R90H mice natively and subsequent to MNV infection, and Fig. 7 demonstrates that TLR7 expression is augmented in R90H mice compared to WT mice, and augmented by MNV infection.*

Overall, this is an interesting manuscript which provides additional insights into the pathogenesis of SLE. However, some data that is presented in this manuscript is not new, and it remains unclear how the SLE-facilitating infection of mutant mice with MNV can serve as a general model for the interaction of genetics with environmental factors.

Granted, this is an interesting example. However, while providing detailed phenotyping of the MNV-infected Ncf1R90H mice, in part seemingly replicating previously published data from the same group, this work does not include experiments with environmental stimuli other than MNV. The authors state at the end that their „discovery of MNV as a trigger of lupus in Ncf1R90H mice does not exclude other viruses, environmental factors or genes, to be causative for SLE“. I consider this a limitation, rather than an asset of this work. The conclusion may be true that this „is the first time a disease-causative interaction has been shown between a defined genetic

and a specific environmental factor to control a complex autoimmune disease”, but the generalizability of this finding remains to be shown.

Authors: Thanks for careful reading and valuable comments on our manuscript. All the data in the manuscript are new as discussed below. It could be but we did not expect it to be a general model with one inducing environmental factor or one genetic factor. It is a complex disease with multiple environmental factors interacting with multiple genes. Now MNV is clarified as one of the environmental factors and NCF190H is one of the genes. To our knowledge, it is the first time it has been shown that an identified SNP interacts with a defined naturally occurring environmental factor to produce a complex autoimmune disease. Just as the reviewer said, we expect additional genes and environmental factors to be discovered, to provide more additional insights into the pathogenesis of SLE.

Major issues

1. *The authors have previously published experiments that are very similar, if not identical to the data shown in Fig. 3 (Li et al., Antioxidants 2023;12:148, Pubmed ID 37507888, ref. 36). Please revise the manuscript so that only novel data is shown, and avoid duplication.*

Authors: We understand the concern raised by the reviewer about the issues of similar data. However, we would like to clarify that all data presented is novel and unpublished. In this manuscript, we show the intracellular and extracellular ROS production in bone marrow derived macrophages (BMDMs) which were obtained from the differentiation of monocytes recovered from the femur and tibia of the Ncf190H mice, and the compensating experiments of ROS production was done after treatment with NOX2 inhibitor GSK2795039 or H₂O₂ (**Fig. 3**). What we have shown previously is a map of ROS production, including intracellular ROS levels in monocytes, neutrophils, and macrophages from peripheral blood, bone marrow, and spleen, also extracellular ROS in PBMCs, bone marrow cells, and splenocytes, and in vivo ROS production, as controlled by the Ncf1R90H polymorphism (Li et al., **Antioxidants** 2023;12:148). The take home message is similar, but the investigations are different with new issues answered. In contrast to previous work, we differentiated bone marrow derived macrophages (BMDMs), testing both intra- and extracellular ROS with and without stimuli, comparing it with the positive

control mouse m1J mutation and we also blocked it with a NOX2 inhibitor or supply with H₂O₂. Although the main take home message can be referred to in a previous paper, we still think it is of value to show here the new and extended data on the ROS production from BMDMs before the further investigation of mechanisms shown in **Fig. 3c-k**.

2. Notwithstanding the suggested revision of Fig. 3, some of the experiments shown in Fig. 3 need some clarification and better controls. For example, in Fig. 3e, Jak1, Stat1, and Stat3 phosphorylation is normalized to b-actin. Without normalization to the unphosphorylated levels of these proteins, it is possible that the effect shown in Fig. 3e is the same as shown in Fig. 3d. Panel e suggests that the phosphorylation (i.e., activation) response to IFN-alpha is exacerbated in mutant mice, but this cannot be asserted without controlling for unphosphorylated levels. (Fig. 1k, on the other hand, does show normalization to total Jak1 and Stat1 protein levels.)

Authors: Thanks for your suggestions. The reason we used the β -actin to normalize against is we aimed to investigate three different proteins. The question was whether the expression of these were increased rather than if phosphorylation of each protein were more pronounced. In a previous paper, we measured expression and phosphorylation of STAT1 (p-STAT1) and found higher STAT1 and p-STAT1 which were normalized to cyclophilin A respectively. It was presented in the diseased kidneys of Ncf1^{m1J} mice compared with WT mice (See **Ref 24**: Hu et al, JCI Insight 2023, Figure 3, E and F). Similarly, our present results show that naïve macrophages from mice with NCF190H allele or Ncf1m1J mutation had a higher level of total JAK1, STAT1, and STAT3 expression than wild-type Ncf1R90 mice. At the same time, IFN- α treatment led to higher levels of phosphorylated molecules. Here we used Ncf1m1J as positive control based on our previous findings. Hope it's good to keep here. The normalized ratio of phosphorylated protein to reference protein β -actin or total protein to reference protein were calculated, and as shown, we did not see any obvious increased ratio of phosphorylated proteins normalized to the unphosphorylated proteins. It's obvious that macrophages from mice with NCF190H allele or Ncf1m1J mutation had a higher level of JAK1, STAT1, and STAT3 expression than wild-type mice from the beginning

naïve status, and of course also activated status. In the peritoneal exudates macrophages (CD45⁺ CD11b⁺ F4/80⁺ Ly6C⁺), we could only detect increased p-STAT1, not increased STAT-1. So, we have added the relative expression of p-STAT1/ STAT1 (**Fig 3j, k**). We hope this clarifies the issues.

3. Please clarify what Fig. 3i adds to the main findings of this work. If the authors deem this panel necessary, they should please consider including non-IFN-alpha as well as wildtype control conditions.

Authors: We agree that Fig 3i does not add anything critical as it is done with the m1j mutation. The reason to show the data from the experimental panel was because we previously reported higher STAT1 and p-STAT1 presented in the kidneys of Ncf1m1J mice compared with WT mice (See **Ref 24**: Luo et al, JCI Insight 2023, Figure 3, E and F). So, we used Ncf1m1J as positive control in the present work and the results are included in both blots and as statistics. We think for completeness it is better to keep the data as it has been done in parallel with the R90H data. Hope it's ok to keep. As for the non-IFN-alpha as well as wildtype control conditions have already been done at the beginning and it has been shown in **Fig 3c and 3d**. So, we did not repeat it here in **Fig 3f** considering the practical experimental design and the limitation of the space. Hope it's more logical and clearer to show.

4. Please clarify the methodology that was used to obtain the data shown in Fig. 3k.

Authors: We highly appreciate your comments on the methodology section, which have helped us to refine our approach. The methodology underlying **Fig 3j and k** is described now in highlighted Results (**Line 249-252**) and Materials and Methods/ Flow cytometry (**Line 729-734**).

5. Fig. 4 seems to repeat and expand on Fig. 1 with the major difference that the mice of Fig. 1 were "naturally" infected, while the mice of Fig. 4 were experimentally and deliberately infected. If this is correct, the two figures should please be consolidated.

Authors: Thanks for your valuable comments. It's correct that the data looks similar, but this is exactly our point as we here show that the effect could be reproduced with an isolated specific MNV strain "Berlin/06/06DE

S99", so it is important and a key to the paper to show both virus strains. Additionally, these two experiments were done separately in different animal facility of China and Sweden, with different experimental aim and design. Experimentally infected with MNV Berlin/06/06DE S99 caused more severe disease as shown in Fig. 4, than natural infection of MNV-59591, shown in Fig. 1. It could be due to the dose and subtypes of virus into the mice, and it was not possible to control the dose infected naturally. Therefore, we isolated and sequenced the MNV strain-59591 naturally infected, which are shown in **Supplementary Table 2** and **Supplementary Fig. 2**, also attached in Figures for review. Most importantly, we just show the simple phenotypes in Fig. 1. We reproduced the phenotypes with isolated MNV and explained further with a strong activation of T and B cells to produce the lupus associated autoantibodies and other phenotypes. Thus, we hope it is agreeable to have and show the experiments separately.

6. All in all, it becomes clear that R90H mice have a propensity to develop an exaggerated im-mune activity/immune response, including the serological response to MNV infection. It re-mains unclear however if this is specific to MNV.

Authors: Thanks for your comments. Actually, we aimed to show that there is an immune response to MNV as well as an autoimmune response. The aim was not to investigate if there are any cross-reactivities in the response to MNV. We have no evidence that the adaptive immune response to MNV per se causes the disease, rather the infection by ssRNA from MNV triggers immune cells into an interferon signaling pathway and into autoimmune responses. How RNA induces an autoimmune response and how it leads to lupus are currently under intense studies by many laboratories, including ours.

7. Could the authors please shed light on how their SPF-housed mice could be accidentally in-fected with MNV in the first place.

Authors: This is in fact an important question as it is likely that many SPF animal houses in fact are infected with viruses which are not pathogenic in normal circumstances (like MNV) as it has relatively recently been identified and has previously not been included in screening protocols for SPF.

However, we have had clean facilities for our animals, both in an animal house in China and in Sweden and it has therefore been possible to identify the effect of the virus. We do not know how the infection occurred in the first room, but it could have been transferred from wild mice or possibly from neighboring animal houses.

Minor issues

1. The keys to colors and symbols should please be placed in a more visible way in Figs. 1 and 2. Presently the key is included for panel B of Fig.1 only. Fig.2 is lacking a key entirely. Although the colors and symbols are evidently used consistently across figures, each figure should stand on its own.

Authors: Thanks for your reminder. It's fixed now. The legends (keys to colors and symbols) have been added into **Fig. 1** and **Fig. 2** across figures.

2. Results section, second sub-heading ("Environmental..."): typo in "arthritis".

Authors: It's now corrected. Thanks for all suggestions improving the manuscript!

Reviewer #2 (Remarks to the Author)

1. *This paper describes the effect of a missense mutation in the NCF1 gene on the immune response to infection with a murine rotavirus. The NCF1 mutation is associated with the risk of SLE although the mechanism by which this occurs has not been firmly established. In this paper, the authors describe the immunological consequences of a usually non-pathological infection which unexpectedly (and fortuitously for the investigators) was found to promote systemic autoimmunity in the mouse carrying the human SLE risk allele. The experiments are thorough and well-presented and support the conclusions drawn.*

Does the missense mutation alter the pattern of tissue expression of NCF1? It is known that interferon can affect NCF1 expressions and hence there is scope for interaction between the mutation, viral infection and qualitative and quantitative aspects of expression.

Authors: Thanks for our great comments. Just a note, we have not addressed rotavirus, only norovirus, which is ssRNA rather than dsRNA. Whether this mutation affects NCF1 expression is an interesting question and not completely understood. The amino acid replacement primarily affects the assembly of the NOX2 complex on membranes, which is likely the main reason for the decreased capacity to mount a ROS response, thus decreasing the ROS response. It has no major effect on expression of NCF1 although it tends to be an effect, please see our previously published data (Li et al., *Antioxidants* 2023;12:148) in which we show that it does not significantly affect expression of NCF1. In fact, it seems to behave very similar to the effect of the SNP identified in the rat, replacing T153M, which is in the same domain (Hultqvist et al *ARS* 10.1089/ars.2010.3440).

Minor points

3) *Abstract: "We have identified a causal SNP in NCF1" is ambiguous in that the most comprehensive paper describing this association was published by others (ref 9) and the discovery of the causal allele had already taken place and is not part of this paper.*

Authors: The statement is correct, but we can rephrase it. The SNP was identified based on the discovery of the rat SNP positioned to be associated with autoimmune disease (Olofsson et al *Nat Gen* 2003) and we

later identified the human SNP by exon sequence (Olsson et al ARS 2012) and subsequently typing in SLE cohorts (Olsson et al ARD 2017). The same year an independent report used the same SNP to type other SLE cohorts (Zhao et al Nat Gen 2017). We have now rephrased the sentence in the abstract (**Line 78-81**).

4) *Typo in title in line 201*

Authors: Thanks for your feedback. Now it's corrected to "Arthritis" (**Line 205**).

5) *Explanation of TLR7-F in Figure 7*

Authors: We have now revised it in the figure legend of Figure 7 (**Line 614-615**) as follows: "Immunoblot analysis of intact TLR7 protein, cleaved TLR7 fragment (TLR7-F)". Thank you once again for your valuable feedback. It has been helpful in improving our manuscript.

Reviewer #3 (Remarks to the Author)

1) Li et al. describe the interesting observation that NCF190H mice develop a lupus-like autoimmune syndrome when infected by murine norovirus, an otherwise not very pathogenic virus. At the same time, the animals clear the virus more quickly, likely due to the exaggerated type 1 IFN response. The paper carefully characterizes the immune response in 90H mice infected either orally (the usual route of infection) or systemically (by IV+IP injection of virus). In both cases, the serum titer of anti-norovirus Abs correlates with the severity of the autoimmunity. This paper is strictly an observational study.

Authors: We are grateful for your valuable input. We would like to point out that the paper also contains conclusive experiments identifying a novel inducing factor (MNV) interacting with the major causative SNP, thus not only observational.

2) It reports an interesting phenomenon that highlights the interplay between genetic risk alleles for autoimmunity (such as NCF190H) and environmental factors, such as viral infection. However, this group has been studying the NCF190H allele and the related NCF1n1J mice for quite a while, in a variety of autoimmune models in a variety of murine genetic backgrounds. Their mechanistic understanding is that reduced expression of the p47phox protein (encoded by the NCF1 gene) leads to decreased ROS production in myeloid cells.

Authors: We are deeply grateful for your recognition of our work. In response to your comments, what we have shown is that polymorphism, in both rats and humans, decreases the capacity to make a ROS response (not primarily expression) and thereby regulating autoimmune diseases. The specific outcome is dependent on the type of mutation in combination with the inducing factor, and this combination triggers different pathways with different mechanisms. The NCF190H allele is recently clarified. Our intention is to investigate the disease triggered by norovirus in mice with human NCF190H allele, with an open mind. And ROS production is reduced but the expression of the p47phox protein (encoded by the NCF1 gene) is intact in our Ncf190H mice, different from the Ncf1m1J mutation.

The initial step, shown in this manuscript, is that the activation of JAK-STAT1 pathways in myeloid cells are involved.

3) This remains a somewhat superficial mechanistic understanding of how the SLE risk allele contributes to autoimmune susceptibility (have the authors ever examined cysteine phosphorylation of specific tyrosine phosphatases such as SHP-1, a known regulator of the Jak/Stat pathway).

Authors: We have not addressed the mechanisms leading to SLE yet. We have now shown the MNV in NCF190H mice leads to a disease mimicking lupus and that it is related to the RNA induced activation of JAK1-STAT1 pathway and that this pathway is regulated by NOX2 derived ROS. There are numerous other intracellular proteins regulated, most likely including SYK but also PTPN22, JAK1, STAT1, STAT3 and LAT based on our published and some unpublished preliminary research. Their relative importance and role in the ROS mediated regulation needs to be determined. In addition, we need to determine how the activation of this pathway leads to pathogenicity. As we earlier have shown in studies on how NCF1 regulates the development of arthritis. Here ROS could operate both extra- and intercellularly. (see Gelderman et al PNAS 2006 and Gelderman JCI 2007).

4) However, that is not the point of the current paper. It is to simply report the cool finding that infection of these mice with norovirus leads to autoimmune disease.

Authors: Thank you! We fully agree. And thanks also for all other valuable suggestions.

Some minor improvements may help with this report:

1). All the data is with 90H mice compared to R90 animals. Do the R90H heterozygous mice develop autoimmunity following norovirus infection (either with or without pristane injection)? I am guessing they do not (or the authors would have told us). One would assume that the majority of SLE patients with the NCF1R90H allele are heterozygous (is this true?), so knowing whether the R90H mice have increased autoimmunity following infection would be important. If they do not, the authors need to put a

paragraph in the Discussion about why a heterozygous risk allele confers disease susceptibility in humans but not in mice.

Authors: Thank you very much for your careful review and valuable comments on our manuscript. Regarding your question about R90H heterozygous mice, we would like to explain that heterozygosity of NCF1 decreases the overall ROS response with all mutations we have studied so far (M153T, m1J, R90H), although the homozygous effect is stronger (Li et al., *Antioxidants* 2023;12:148). Clearly there is also an impact on disease in humans as well as in animals. However, a larger number of animals are needed to obtain enough statistical power, making it difficult to include in all experiments. As you know, in some of the early experiments we did include heterozygous mice from our littermates breeding. Just to clarify, all the wildtype, heterozygous and homozygous B6NQ mice were infected with MNV and followed either without (Month 0) or with injection of pristane (Month 6 and 7). It only shows a significance between wildtype (R90) and heterozygous (R90H) mice with arthritis scores on the six and seven month-injection of pristane together with infection of MNV in female mice (Fig b). It also only shows weakly significant, with very low levels of anti-dsDNA antibodies between MNV infected both female and male wildtype and heterozygous mice before/ without pristane (Month 0). Arthritis did not develop in the female B6 background mice. Significance of anti-dsDNA antibodies was shown in both heterozygous and homozygous female B6NQ mice compared with wildtypes, which gave us the insight that environmental MNV alone can also induce lupus in the R90H or 90H mice. To avoid large number of mice in the experiments, we did not regularly use the heterozygotes in the experiment. However, since this question was raised, we now decided to show this data as **Supplementary Fig. 6** (also attached in Figures for review). But please note that some data is repeating with Fig 2.

2). Where are the viral titers for the oral infection data reported in Fig. 1? They are reported for Fig. 4.

Authors: We appreciate your valuable advice. In fact, we detected the virus RNA with environmental MNV infection but with Ct value. Nevertheless, we suggest adding the data into **Supplementary Fig. 3a**, also attached in Figures for review.

3). *Fig. 1 and Fig. 4 both report outcomes with oral infection by norovirus - Fig. 1 by exposure to infected feces and Fig. 4 by oral gavage. The analysis is a bit different for each figure. Is there any reason to think there would be a difference between these two methods of infection? The data seemed a bit redundant.*

Authors: We are deeply grateful for your insightful feedback. As explained above, it's an important point that we could reproduce the environmental induction with an isolated strain. So, we are happy that the results look similar, and we would like to show this as it's the essential cornerstone of this paper. However, there are differences as the mice experimentally infected with MNV caused much more severe diseases as shown in **Fig. 4** than the natural infection shown in **Fig. 1**. We have no obvious explanation for this difference, but it could be due to the dose and subtypes of virus into the mice, as well as the physical environmental conditions. The experiments were done separately in different animal facilities in China and Sweden and with different laboratory possibilities. Therefore, we isolated and sequenced the MNV from the feces used for natural infection. For injection experiments, we used the MNV strain Berlin/06/06DE S99. Most importantly, we show the simple phenotypes in Fig. 1. We repeated the phenotype with isolated MNV and explained more with a strong activation of T and B cells to produce the lupus associated autoantibodies and other phenotypes. Hope it's more logical and clearer to the readers.

4). *It wasn't quite clear, but does oral infection by norovirus lead to the anti-COL1 IgG/IgMs reported in the IV+IP model in Fig. 5? It is somewhat difficult to interpret the different immune response analyses reported in the different models. It would seem more logical that the same analysis be completely done on mice infected in different ways.*

Authors: In response to your concern, we need to clarify that we did measure antibodies to COL2 IgG/ IgMs antibodies also in mice infected per oral with MNV. However, the anti-COL2 titers were very low. Keep in mind however, that the mice infected perorally with MNV did not develop arthritis and therefore, the response to COL2 could be a secondary immune response to the arthritis but it could also be due to other reasons. It could also be the other way that the anti-COL2 antibodies caused

arthritis, although this is not likely as the titers is too low to be arthritogenic. In BALB/c.90H mice with environmental MNV infection, we observed lupus arthritic paws with histological symptoms, but it seems to be different from the arthritis we observed in the IV+IP model. Please check the data (**Fig. R1** in Figures for review attached). But we suggest not including it in the manuscript as we are not yet ready to provide a mechanistic explanation for the difference.

5). It wasn't quite clear if the pristane + norovirus-infected mice developed a more severe form of autoimmunity than the norovirus infected alone. Fig. 2 only compares pristane against pristane + norovirus. The authors may wish to comment on this.

Authors: Thanks for your valuable comments. But we need to clarify that we aimed to investigate the role of MNV in pristane-induced lupus (PIL) model instead of the role of pristane, so we kept pristane injected groups but one is with MNV infection, another is without MNV infection. Based on the data, we concluded that the lupus severity was significantly aggravated in the 90H mice once/ only infected with MNV. We even detected weak but statistically significant autoantibodies in 90H mice at the early time point around 35 days of MNV infection but before injection of pristane. It was completely consistent with the findings we clarified. That's what we really want to know in this manuscript. We don't know if the pristane + norovirus-infected mice developed a more severe form of autoimmunity than the norovirus infected alone, without norovirus alone group as parallel. Of course, we agree that we can compare pristane+ norovirus with norovirus alone so that we know the role of pristane in MNV- induced lupus. But it seems too complex to clarify in this condition. Because it seems to be different mechanisms with pristane injection and with MNV infection. MNV-induced lupus is more NCF1- dependent than PIL model. That's for sure we can investigate more but that cannot be in this manuscript. We would like to investigate it in the later project or active readers/ researchers to do more in the future. Taken together, we simplified **Fig 2d** so that it is easier to understand but kept the complete data in **Supplementary Fig. 6**, also attached in Figures for review. Thanks for your valuable insight and improvements to our manuscript.

6) I think Fig. S10A needs a legend. Which are the Yaa-containing mice? The authors don't comment on whether breeding the Yaa allele into the 90H background (as homozygotes) leads to more severe disease than Yaa alone. One would think that it does.

Authors: The figure legend of Fig. S10A (Now is **Fig S11a**) has now been added as "The Yaa-carrying mice strain with fully functional NCF1 (R90.Yaa) or with human NCF190H allele (90H.Yaa) was obtained by crossing homozygous BQ.Ncf190H/90H with B10.Q.Yaa. And heterozygous BQ.Ncf1R90/90H.Yaa were intercrossed for the experimental mice." We have previously reported that NCF1-dependent ROS deficiency drives spontaneous lupus development in mouse mutant Ncf1^{m1j} mice carrying Yaa (JCI Insight 2023;8(7):e164875). At the beginning, we also expected spontaneous lupus in Ncf190H.Yaa mice. So, we had set up some preliminary experiments with 90H.Yaa mice. We could observe a trend but no significance regarding levels of proteinuria, anti-dsDNA antibodies or anti-Sm/RNP antibodies between 90H.Yaa mice and Yaa alone (R90H.Yaa), or between 90H.Yaa mice and 90H mice. Here we followed the mice for 6 months (**Fig. R2a** in Figures for review attached). Or even followed the mice for 12 months (**Fig. R2b** in Figures for review attached). And we also compared the lupus symptoms between 90H.Yaa mice with R90.Yaa mice 2-month post of MNV infection in a natural way. We did not see the difference either (**Fig. R2c** in Figures for review attached). It could be that Yaa locus alone (R90.Yaa) play role in developing lupus, and it covered the role of 90H allele when it's together with Yaa locus.

7). For the one sentence about autoimmunity and COVID19 in the Introduction, the authors should check out this very recent report, PMID: 39112696

Authors: It's a good paper to cite in our manuscript as ref 19 (**Line 146**). It greatly strengthened our point. And it also shows a good insight into further research.

Overall, this is an interesting phenomenological report. The authors must have been really surprised when, some years ago, their colony of 90H mice

suddenly started developing arthritis! They have done a great job figuring out why.

Authors: We appreciate the reviewer's time and effort in providing these comments. We hope that our revisions have addressed all the concerns raised. Thank you once again for your valuable feedback. It has been extremely helpful in improving our manuscript.

Reviewer #4 (Norovirus, mucosal immunity) (Remarks to the Author)

Li et al demonstrated that infection of MNV with the NCF190H mice, whose SNP causing SLE in humans, induces lupus. The authors also found NCF190H allele upregulated INF- α /JAK1/STAT1 pathway-enhancing type I interferons and antibodies against MNV, and TLR7 in hemopoietic cells being activated by mucosal MNV infection. Taken together, the authors concluded that NCF190H is sensitive to MNV infection, resulting in protection against MNV infection and the development of murine lupus. This reviewer thinks this manuscript is important to understand the mechanism of SLE development. In particular, identifying MNV as an environmental factor to induces lupus is outstanding, and expect to apply it for the establishment of therapy in humans.

Authors: We are grateful for your summary and recognition of our work.

To understand this study deeply, this reviewer expect to add description to answer some questions as below.1. Although the authors concluded that ssRNA virus infection triggers the development of lupus, the conclusion is a little bit strong to say by only one MNV strain. Have you tested minimal other MNV strains because there is a major distinct pathogenesis, acute and chronic?

Authors: Thanks for your valuable comments. Actually, we show data from two different MNV strains. One is a natural environmental MNV strain 59591 studied in Fig. 1. After discovering the importance of the natural infection, we sequenced and studied the phylogenetic relationships between MNV isolate-59591 with other known MNV. Nucleotide sequences were compared for the whole genome (**Supplementary Table 2/ Isolate 59591 and Supplementary Fig. 2/ Query_59591 in**). MNV strain used for acute and chronic infection is Berlin/06/06DE S99 (**Fig. 4**). We observed the same lupus phenotype in these two specific MNV strains. We have specifically clarified both of them in the main text now. Of course, we could also test another MNV strain like MNV-1 strain commonly used. And we can also compare MNV-1 with Berlin/06/06DE S99. But this is not our aim in this study.

2. Does this author know the allele affects functional change in the intestine? Does the expression of MNV receptor proteins change? Although data on infection of MNV by i.g. might support this question, this reviewer would like to know the detailed mechanism in the intestine, which is not fully covered yet in this manuscript.

Authors: Thanks for your insightful questions. **1)** Just as reviewer mentioned, we have data describing the response to MNV i.g. infection. It shows the Ncf190H allele, together with MNV gastrointestinal infection both in the Peyer's patches (PPs) of intestine and systemically affected T cell-dependent germinal center formation and memory T cell responses, lead to a strong antibody response. We also observed dramatically decreased ROS mainly in the intestines of both homozygous (90H/90H) and heterozygous (R90/90H) mice, even after LPS stimulation, but not in the lung, spleen, liver, heart or kidney, when compared with wild-type R90/R90 mice, as detected using the IVIS 50 bioluminescent system. Our data proved this functional change in the intestine is completely due to Ncf190H allele in the naive status, which was previously published in the paper "Li et al., *Antioxidants* 2023;12:148" (See **Figure R3/** Li et al. Published data *Antioxidants* 2023 attached in Figures for review below). **2)** It's rarely reported that murine norovirus is recognized by CD300lf receptor that mediates binding to the cell surface and by the pattern recognition receptor MDA5 in the cytoplasm (Orchard RC et al., *Science*. 2016, PMID: 27540007 & Yu P et al., *Gut Microbes*. 2021, PMID: 34347572, Fig S2, Fig 2 (e, f)). STING agonist DMXAA treatment of mouse macrophages increased the expression and phosphorylation of STAT1, and the transcription and protein expression of several antiviral ISGs, including the innate immune sensor MDA5. Moreover, treatment with JAK inhibitor partially attenuated DMXAA-mediated anti-MNV ability. Considering the aim of our study, we did not investigate the direct effect of the NCF1^{90H} allele on the expression of MNV receptor. Indirectly, we investigated the role of NCF1^{90H} allele on DMXAA-mediated anti-MNV ability in bone marrow derived macrophages stimulated with STING agonist DMXAA, or both of DMXAA and JAK-inhibitor, which were more related to our study. Here we proved that NCF1^{90H} upregulates p-STAT1/STAT1 through the activation of STING. While JAK-inhibitor can downregulate it (**Fig R4 Exp 1 & 2** in Figures for review). We know RIG-I, cGAS and STING can restrict

MNV replication in mouse macrophages, which could be involved in MNV recognition, but it needs further study. In the context of MNV infection, we encourage future studies to decipher the potential crosstalk between the CD300lf/MDA5-MAVS and cGAS-STING axis.

3. Which region does the sequences in Table 2 show?

Authors: Thanks for pointing it out. It's the region from bases 5027 to 5078 (Genomic RNA is from bases 1 to 7382). It has been added into the legend of **Supplementary Table 2**, also attached in Figures for review.

4. In Fig S1, the description of p-values isn't correct.

Authors: Thanks for reminding us. It's corrected accordingly and revised in full manuscript.

5. The name of the MNV isolate could be unified among Fig.S3 and Table S2.

Authors: Thanks for your great suggestions. It has been unified as Isolate-59591. Hope it looks better.

8. In Fig 1h, the right panel showing IgG and C2 in MNV-90H seems to be too bright rather than the left images. Is this the correct image? On the other hand, the image, Ncf1m1J (? , unreadable due to low resolution) in Fig S7c, seems to be too dark. These images could not support descriptions for each.

Authors: Thanks for your valuable feedback. It's corrected with the images in Fig 1h and Fig S7c accordingly, and also figure legend in Fig S7c. Hope it looks better now. The results of western blot in Fig 3 could be more supportive than immunofluorescence staining in Fig S7c. But still think it's good to keep immunofluorescence staining as supplementary materials. Thanks for your help to improve our work!

Figures for review

Table S2: Major MNV nucleotide sequences.

	Score	Expect	Identities	Gaps	Strand
	97.1 bits (52)	7e-17	52/52 (100%)	0/52(0%)	
	Plus/Plus				
Isolate-59591	CCCGCAGGAACGCTCAGCAGTCTTTGTGAATGAGGATGAGTGATGG CGCAGC				
CW3	CCCGCAGGAACGCTCAGCAGTCTTTGTGAATGAGGATGAGTGATGG CGCAGC				
MNVSH1603	CCCGCAGGAACGCTCAGCAGTCTTTGTGAATGAGGATGAGTGATGG CGCAGC				
HBTS-1806	CCCGCAGGAACGCTCAGCAGTCTTTGTGAATGAGGATGAGTGATGG CGCAGC				
BJ 10-2062	CCCGCAGGAACGCTCAGCAGTCTTTGTGAATGAGGATGAGTGATGG CGCAGC				

Fig. S2 Phylogenetic relationships between MNV isolates with other known MNVs. Nucleotide sequences were compared for the whole genome.

Supplementary Fig. 6

Supplementary Fig. 6 MNV aggravates pristane-induced lupus in BQ.

***Ncf1*^{90H}** mice. Mean arthritis, anti-dsDNA antibodies and anti-MNV antibodies in BQ. *Ncf1*^{90H} male mice (n = 4-8). **a** and female mice (n = 9-11). **b**. Anti-dsDNA antibodies and anti-MNV antibodies in Female B6 background mice (n = 7-16). **c**. Arthritis did not develop in the female B6 background mice. Data shown are mean ± SEM. R90 vs 90H: * $p < 0.05$; R90 vs R90H: ^ $p < 0.05$.

Supplementary Fig. 3a

Supplementary Fig. 3a Environmental MNV induces lupus in BALB/c.*Ncf1*^{90H} mice. Virus RNA were detected from the feces collected on day 7 after exposed into the environmental MNV. RT-qPCR was performed to assess MNV load with Ct value.

Fig. R1

Fig. R1 Mucosal MNV infection induces lupus in BQ. Ncf190H mice. The level of anti-collagen II (COL2) IgG antibodies, anti-COL2 IgG2b antibodies, and anti-COL2 IgM antibodies in sera of mice.

Fig. R2

Fig. R2 The role of Yaa locus in lupus. a. The level of proteinuria, anti-dsDNA antibodies or anti-Sm/RNP antibodies in mice with 90H, 90H.Yaa and Yaa alone (R90H. Yaa), or between 90H.Yaa mice and 90H mice, which followed for 6 months. **b.** The level of proteinuria, and anti-dsDNA antibodies in mice with Yaa alone (R90H.Yaa) and 90H.Yaa, which followed

for 12 months. **c.** The level of anti-dsDNA antibodies and anti-ssRNA antibodies in mice with Yaa alone (R90H.Yaa) and 90H.Yaa, which were naïve (without MNV infection) or 2-month post of MNV infection in a natural way.

Fig. R3 (Li et al Published data Antioxidants 2023)

Fig. R3 NCF190H variant decreases extracellular ROS and in vivo ROS production. Decreased ROS in vivo and the dissected organs by ex vivo imaging, injected with L-012 probe after LPS stimulation for five hours. Quantification of in vivo generated ROS (n = 5–7).

Fig. R4

Fig. R4 The role of NCF190H allele in STING/p-STAT1/STAT1 axis. Bone marrow macrophages stimulated with STING Agonist DMXAA, and both DMXAA and JAK-inhibitor.

Round 2

REVIEWERS' COMMENTS

Reviewer #1 (Remarks to the Author):

Thank you very much for the amendments made to the manuscript. I still contend that part of this work provides only incremental insights over the Li 2023 paper, but this shall not hinder publication.

Authors: Thanks for reviewer's valuable comments and approval of our work.

Reviewer #3 (Remarks to the Author):

The authors have made a number of clarifications in the manuscript in response to suggestions from the reviewers. Most of the reviewers felt this was an interesting story. I have no further suggestions.

Authors: Thanks for your approval to our work.

Reviewer #4 (Remarks to the Author):

This reviewer agrees with the current manuscript including revisions to respond to reviewer comments properly. However, this reviewer would like to suggest a comment related to question 1, which could strengthen this manuscript.

This reviewer agrees with the conclusions which are drawn from data by engaging two MNV strains, Isolate 59591 and Berlin/06/06DE S99. However, these strains are phylogenetically classified comparatively close rather than MNV-1, that is, seem to show similar pathogenicity. Question 1, which this reviewer would like to ask, was if both MNVs, which cause different pathogenesis, acute and chronic infection, result in the same phenotype. This reviewer guesses the answer is NO because the different infection style leads to distinct immune responses. This reviewer accepts the author's claim that testing another MNV strain like the MNV-1 strain is out of aim. On the other hand, this reviewer would like the authors to discuss the possibility of

differences in lupus induction between the strains used in this study and MNV-1 to avoid misunderstanding readers that all MNV strains cause the same phenotype.

Authors: Thanks for reviewer's suggestions. We agree with reviewer and have made corrections accordingly. We will also investigate the role of different strains further in our forthcoming studies.